# Equality in Income and Sustainability in Economic Growth: Agent-Based Simulations on OECD Data

**Shungo Sakaki**

School of Media Science, Tokyo University of Technology, 1404-1 Katakura, Hachioji, Tokyo 192-0914, Japan; sakaki-s@y4.dion.ne.jp; Tel.: +81-42-637-2785

**Abstract:** In countries that have developed under the current market economy, inequalities in income distribution tend to increase with three different trends, i.e., high (United States, United Kingdom, Japan), low (North Europe countries), and medium Gini coefficient levels. On the other hand, the relationship between income distribution and social welfare is generally a difficult problem to solve in economics. So, this paper discusses the impact of income distribution on the macroeconomy, limiting the scope to consistency with long-term economic growth. We attempt to answer these economic policy issues by simulation using an agent-based model based on replicator dynamics. As a result of the simulation in this paper, in general, in countries with the high marginal propensity to consume, long-term growth can be maintained by inducing equality in income distribution. On the other hand, a mature country with a low marginal propensity to consume can sustain not so high but stable growth despite increasing inequality in income distribution. According to simulation results based on OECD (Organisation for Economic Co-operation and Development) data, in the former UK, US, and Japan, the lower the Gini coefficient is, the higher the growth potential is, while in the latter Norway and Luxembourg, relatively stable growth is maintained even if the Gini coefficient increases.

**Keywords:** equality in income; sustainable long-term economic growth; marginal propensity to consume; inter-system control; agent-based simulation; Gini coefficient

---

## 1. Introduction

This chapter reviews theoretical studies on economic growth theory and empirical and practical studies on fiscal issues related to income distribution policies, from among a wealth of studies on macroeconomics. Concerning the methodology on model construction and data analysis of agent-based simulation, this paper reviews various literature mainly related to socioeconomic problems from the enormous research results in the same field developed since the 1980s. This chapter positions this paper in the genealogy of the above research area and clarifies the difference and uniqueness.

This chapter begins with an overview of both global trends and country-specific trends in income disparities at the nation-state levels, together with an overview of trends at the industrial and corporate levels in Japan related to income disparities (Section 1.1). Then, we will review existing studies on economic growth and income distribution (Section 1.2) from the viewpoint of economics and existing studies on agent-based models (Section 1.3) from the viewpoint of methodology, position the simulation model adopted in this paper in the genealogy of these existing studies, and examine its validity.

### 1.1. Global Trends and Country-Specific Characteristics of Growing Income Inequality

In countries that have developed under the current market economy, while with low economic growth, inequalities in income distribution tend to increase with three different trends, i.e., high (US, UK, Japan), low (North Europe countries), and medium Gini coefficient levels (Figure 1). Especially in

the Japanese economy, a significant decline in labor's share has been occurring both at the industrial and enterprise levels (Figure 2).

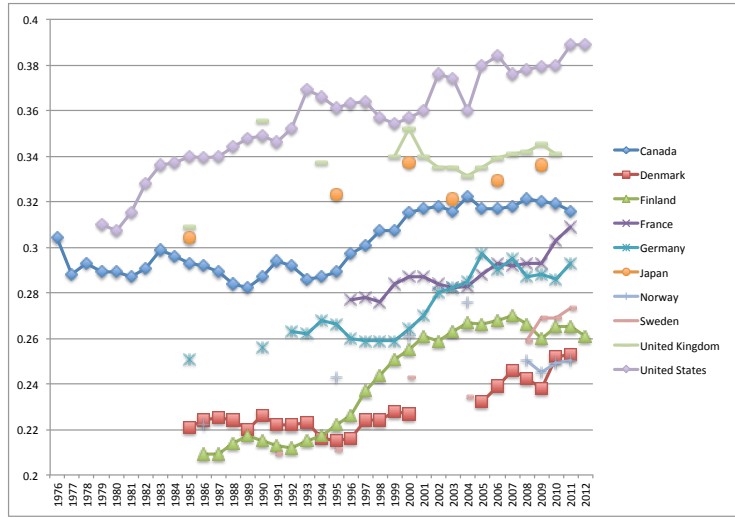

**Figure 1.** The diagram shows the transition of the Gini coefficient in several countries of the OECD (Organisation for Economic Co-operation and Development). Source: OECD.Stat.

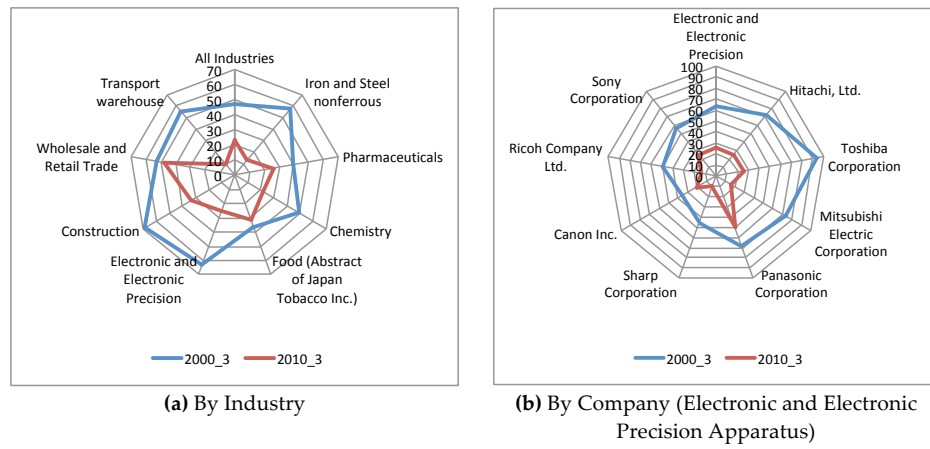

| **(a)** By Industry | **(b)** By Company (Electronic and Electronic Precision Apparatus) |
|---|---|

**Figure 2.** The diagrams show changes in labor's share in Japan. Source: TOYO KEIZAI Inc. "financial chart".

*1.2. Economic Growth and Income Distribution*

There have been several economic theoretical [1,2], empirical [3], law, and political [4–6] studies on the relationship between income distribution and economic welfare as a matter of social choice, but the relation is an extremely difficult problem to solve comprehensively. Therefore, this paper discusses the impact of income distribution on the macroeconomy, limiting the scope to consistency with long-term economic growth. In this section, based on the methodological relevance to the model in this paper, the following is a summary of the genealogy of existing research on economic growth theory and income distribution.

Economic growth and income distribution are the most important issues in economic policy. Economic growth is an area in which many economists have attempted prescriptions in the framework of relatively pure economic theory as a long-term resource allocation problem aimed at sustainability. The theoretical work, starting with Ramsey [7], was concluded by Koopmans [8], Cass [9], and Uzawa [10] as a problem for deriving the optimal growth path that defines the balance between consumption and savings. In addition, based on the theoretical requirement that consumption plans be drawn up over a finite period on the basis of realistic behavioral hypotheses, an overlapping generation

(OLG) model was developed starting with Samuelson [11] and extended by Diamond [12]. In the OLG model, the generations responsible for consumption and savings plans of each generation are inherited from the previous generation and passed on to the next generation, and macroeconomics was refined [13,14].

Mathematical methods for analyzing economic growth based on the framework of the optimal growth model have developed into monetary growth models and endogenous growth models. Romer's endogenous growth model [15] explicitly captures the relationship between technological innovation and economic growth. The endogenous growth model is an optimal growth planning model based on the accumulation of knowledge capital. At the heart of this model is the Ramsey optimal growth model, which incorporates the Marshall externalities as a property of knowledge capital [16].

The agent-based simulation model in this paper, like the conventional economic growth theory described above, consists of a basic macroeconomic model framework. However, unlike the conventional economic growth model, it has explicit sectors based on national accounts [17]. In addition, it explicitly incorporates elements of technological innovation, which are the driving force of economic growth, and forms relationships among sectors based on the framework of industrial organization theory that has reflected the results of game theory since the 1980s [18–21]. However, the model in this paper is not based on the Marshall externalities adopted by Romer [15,16], but rather on an economic growth model incorporating the coordination effect of Cooper and John [22], targeting sustainable growth through the accumulation of knowledge capital.

On the other hand, income distribution is inseparable from social policy issues that have a significant impact on the formation and maintenance of social order. In addition, even if the code of conduct is the main subject, intervention by the authorities on individual values also occurs, and treating it as an essentially pure economic problem itself is a difficult problem to solve. As for income distribution policies, there are theoretical studies on income distribution and economic welfare that started with Arrow and Sen, but it is impossible to solve the general solution of social choice under the general equilibrium theory [1,2]. Turning to empirical research, while there are extensive studies, such as Piketty [3], that measure economic realities within the historical context, individual studies that focus on measuring realities [23–25] or that study issues and measures within the institutional context specific to each country [4–6,26–28], either from the perspective of fiscal authorities also exist.

As outlined above, while the theory of economic growth has a rich lineage of theoretical research, the issue of income distribution is essentially a social choice that goes beyond purely economic policy issues. While the issues of economic growth and income distribution have developed independently of each other, research on the relationship between these two issues, including theories and empirical evidence, has scarcely provided working and practical policy options except some research [29–31].

This paper constructs a simulation model based on the sector/transaction structure of the national accounts and input-output tables, and attempts to present policy scenarios based on actual data. At the end of this section, I would like to introduce a typical research framework that directly links to economic policy based on theoretical research and actual data in this area.

Various studies based on national accounts and input-output tables exist as a representative framework in which economic policy authorities, such as administrative organs, as providers of statistics or implementers of statistics, can conduct consistent research on theories and empirical evidence [32–34].

The system of national accounts is a systematic record of the overall economic activity of a country in an internationally comparable manner in accordance with international standards set by the United Nations and in conformity with national statistics of each country. In Japan, the national accounts are estimated by the Economic and Social Research Institute of the Cabinet Office [35].

On the other hand, the input-output tables, developed by Leontief [36], become compiled into a matrix of how goods and services are produced and sold between the various industrial sectors over a certain period (typically 1 year). In Japan, the tables are prepared every 5 years through joint efforts by

10 government ministries and agencies under the coordination of the Ministry of Internal Affairs and Communications [37].

The national accounts and input-output tables are, in principle, based on the measurement of actual economic transactions based on accounting. As a result, although there are various limitations, they provide an exceptional analytical framework in social science, in which the theoretical structure of the national economy and feasibility of verification are linked inextricably. With regard to economic growth, it is possible to prepare growth accounting from the national accounts, estimate the contribution of economic growth by a production factor, and estimate total factor productivity (TFP) [38].

The input-output table is a vast amount of statistical information that not only expresses the interdependence of goods and services in a country and analyzes the macroeconomic spillover effects derived from the input-output structure but also provides employment and fixed capital matrices, etc., enabling a variety of analyses [38]. However, this industrial structure is fixed, and the input-output table has limitations in the analysis of long-term economic growth and does not provide a direct analytical framework for the issue of income distribution.

Based on the genealogy of research centered on policy theory on economic growth and income distribution, this paper constructs a model in which we can manipulate the relationship between economic growth and income distribution by limiting its range to the relationship with sustainable economic growth. In other words, this paper provides a framework that enables the presentation of policy options, not only in pure theory but also in empirical terms, beyond the measurement of actual conditions. Therefore, to avoid the limitations of theoretical models, we devise an economic growth model based on an agent-based simulation that has not been widely adopted in economics and attempt to provide methodology and policy recommendations that can contribute to economic policy practice.

### 1.3. Why Agent-Based Simulations?

Economics has traditionally oriented toward the application of natural scientific methodologies that are equivalent to both theories and experiments typical of physics. It goes without saying that Smith, Walras, and Keynes were exceptional examples in which a unified theoretical system was established to elucidate socioeconomic phenomena. Friedman has argued that the novelty of policy ensures the validity of the econometric model used to verify the theory [39].

A framework based on natural scientific methodology, in which economic policy authorities construct economic policy options by conducting numerical calculations using econometric models based on orthodox macroeconomic theory, has functioned during the high-growth period after World War II. Since the 1980s, as seen in a number of studies that have focused on game theory or dynamic optimization, it can be said that efforts have been made to reconcile both simplifications with the possibility of mathematical analysis and concreteness with realistic policy implications in specific targeted areas, such as comparative institutional analysis [40–42], strategic trade negotiations [43], oligopolistic markets [44], technological innovation [45], and monetary theory [46,47].

However, it is impossible to directly apply the typical natural science methodology of modern physics, that is, the method of positive proof based on theory and experiment, to social science itself, especially policy issues based on social science knowledge [48]. This is because, even in domestic relations, the socio-economy subject to research itself is a multi-entity complex consisting of individuals; companies; regions; various organizations, such as NPOs (Non-Profit Organizations) and NGOs(Non-Governmental Organizations); administrative organizations, etc. [49], and is composed of complex relationships among such interested parties, thus the model structure itself can be extremely complicated. Besides, there is no doubt that social experiments to test the pros and cons of economic policies are not possible as a process of implementation based on consensus among the people and countries [50].

In principle, it is impossible to provide a socio-scientific methodology that can present realistic policy issues to a socio-economy with a historical, institutional, and cultural context specific to each country and region, as well as a complex relationship among them, based on a conventional model

structure that conducts quantitative verification based on a highly simplified mathematical model that can be analyzed [48]. In addition, theoretical models that specialize in specific areas or even theoretically supported input-output tables have limitations in providing multiple, diverse, substitutable, and long-range scenarios for economic policy options.

As a methodology for coping with the complexity caused by the multi-agent and interdependence of actual social organization and phenomena in the social science field where the application of such analytical methods is difficult, simulation as a computer experiment has been presented mainly from the field of engineering [48]. For example, even in the domain of simple game theory, the "tit-for-tat strategy" in Axelrod's experiment [51] is not a stable cooperative solution of game theory (not sub-game perfect equilibrium) [52], and although it is too simple as a system design for real complex interests, it is still epoch-making in the sense that it opened the way for practical use.

What is an agent-based model that can provide policy options in a complex society? Based on the above points, the following parts introduce the definition and requirements below.

First, society, organization, and individuals are composed of agents in social simulation using agents, and we can examine the process of constructing the system from the bottom up and the nature of the structure through competition and cooperation between them. The agents in the agent-based model have an internal state and decision-making and problem-solving abilities. They act and adapt autonomously, and engage in information exchange and problem-solving. At this time, through the interaction between agents, the macroscopic nature of the targeted system emerges, the agent and the environment surrounding the agent form a micro/macro link, and the state of the system changes while mutually influencing each other [50,53].

The simulation data generated in this way are not obtained by measuring the real world but are generated from the above rules on the decision-making structure and the reciprocal transactions, and can be analyzed inductively instead of examining the correctness through mathematical analysis [54]. In other words, it is an attempt to generate data for the sake of sharing accurate knowledge among researchers on phenomena where the complexity of reality exceeds the possibility of mathematical analysis, and the reproducibility and social experiments are impossible [48,55].

In general, the objective of economic theory is to assume a rational economic entity, to assume the existence of an optimized equilibrium, and to determine this. On the other hand, in the agent-based simulation, it is not necessary to assume the above rationality in the principle of decision-making of the economic agent, and it is possible to directly describe and analyze a boundedly rational agent [56].

Moreover, it is also possible to assume a practical decision-making structure that reflects actual economic transactions beyond bounded rationality. In fact, the agent-based model based on replicator dynamics presented in this paper is based on adaptive decision-making and is consistent with a practical PDCA (Plan, Do, Check, Action) cycle [57]. From the generated simulation data, there is a constraint that it is only a trial calculation result and not a mathematical proof, but through the calculated data, not only the result of the optimum equilibrium state but also the whole calculated process, such as the state of dynamic stability, permanent instability, etc., other than the equilibrium state can be analyzed [56].

In addition, in the process of reviewing the results of simulations, agent-based models can have institutional design objectives in themselves to deepen the understanding of the basic processes that appear in various applications and can propose a framework for problem-solving that can trace various institutions back to the design process [50,53]. In other words, regarding the characteristics of the path (time series data) obtained from simulation, a methodology with an evolutionary process is provided, which does not exist in the conventional "theory → analysis → demonstration" process, but which has the element of PDCA in the agent-based model itself, i.e., "policy scenario review → revision → re-simulation".

In general, there are no restrictions on the theoretical foundations that can be employed to construct agent-based models, and they are implemented in a variety of ways, such as game theory, genetic algorithms, or a mixture of machine and human agents, depending on the targeted application

area [53]. The North American Association of Computational Social and Organizational Science (NAACSOS) in the United States, the European Social Simulation Association (ESSA) in Europe, and the Pacific Association for Agent-Based Approach in Social Sciences (PAAA) in Asia, led by Japan, have yielded many results [53]. On the other hand, the agent-based model using replicator dynamics employed in this paper is a very simple and classical dynamical system but has not yet had many applications [57–61].

Based on the above definitions and requirements, this paper defines R & D (Research and Development), production, and policy sectors as agents; defines their decision-making structures and inter-business relationships in a realistic manner regardless of whether or not they are mathematically analyzable; and constructs an agent-based model that controls policy variables relating to inter-business relationships and income distribution among the sectors based on replicator dynamics. Replicator dynamics can describe state transitions. In the replicator dynamics, the agents are making decisions adaptively as in the PDCA cycle. The PDCA cycle is a practical decision-making method by actual enterprises, in which they revise and execute plans sequentially while comparing plans and results.

This paper bases the structure of the model on the author's previous study [59], but it is a simplified version in which the influence of the income distribution system on economic growth is directly within its range. As a result of the simulation, macroeconomic conditions that are related to income distribution and economic growth emerge from the interrelationships among the various sectors, and inductively analyzable data can be generated on policy variables, such as income distribution rates, consistent with long-term economic growth. In addition to generating data based on a hypothetical scenario, this paper sets the Gini coefficient, population, and marginal propensity to consume based on macroeconomic statistics of OECD countries as a premise for conducting simulations, and analyzes the differences in growth paths of each country. Besides, as a simulation trial, we examined the growth potential of each country and the stability of its path not only based on predictions based on these actually measured variables but also based on assumed values that were impossible as a social experiment.

## 2. Overview of the Model Structure and Procedures for Modeling and Conducting Simulations

In this chapter, after explaining the basic structure of the agent-based economic model in this paper (Figure 3), we will provide a step-by-step explanation of how to construct the agent-based economic model described in Sections 3 and 4 and the procedure for conducting simulations described in Section 5, which are the core of this study, referring to the overall structure of the model (Figure 3). While the overall model may appear complex, the breakdown of the sector's function and the base profit structure for decision-making in each of the following stages consists of very simple elements, such as the productivity of knowledge stocks as defined in the R & D sector (see expressions 6 and 7) and the aggregate consumption demand (see expressions 25, 23, 27, and 21) for each income distribution scheme as defined in the consumer goods production sector.

This paper presents recommendations on policy issues related to income distribution and long-term economic growth using agent-based simulations. Macroeconomics has a complex structure that arises from a large number of factors and the interrelationships between them. Traditionally, both theoretical and empirical studies on this issue, especially those that directly relate them to each other, are still in their infancy (see Section 1). As a framework for dealing with such complex macroeconomic growth, this paper proposes a modeling technique that applies the agent-based social systems science (ABSSS) method [53,60] instead of traditional mathematical analytical economic models while conforming to basic macroeconomic models and also proposes policy options obtained from simulations.

The basic structure of the model (Figure 3) consists of consumption and capital investment on the demand side under national accounts [17,35], and reflects the results of sustained technological innovation by research institutes as factors for productivity improvement on the supply side. The agents of the model are composed of three sectors: The R & D sector as a research institute, the private business sector consisting of the consumer goods production division and the investment goods production division, and the system control sector as a policy authority. In this case, the difference in the

income distribution system directly affects consumer demand by decreasing the marginal propensity to consume and indirectly affects the demand for capital investment through savings-based loans.

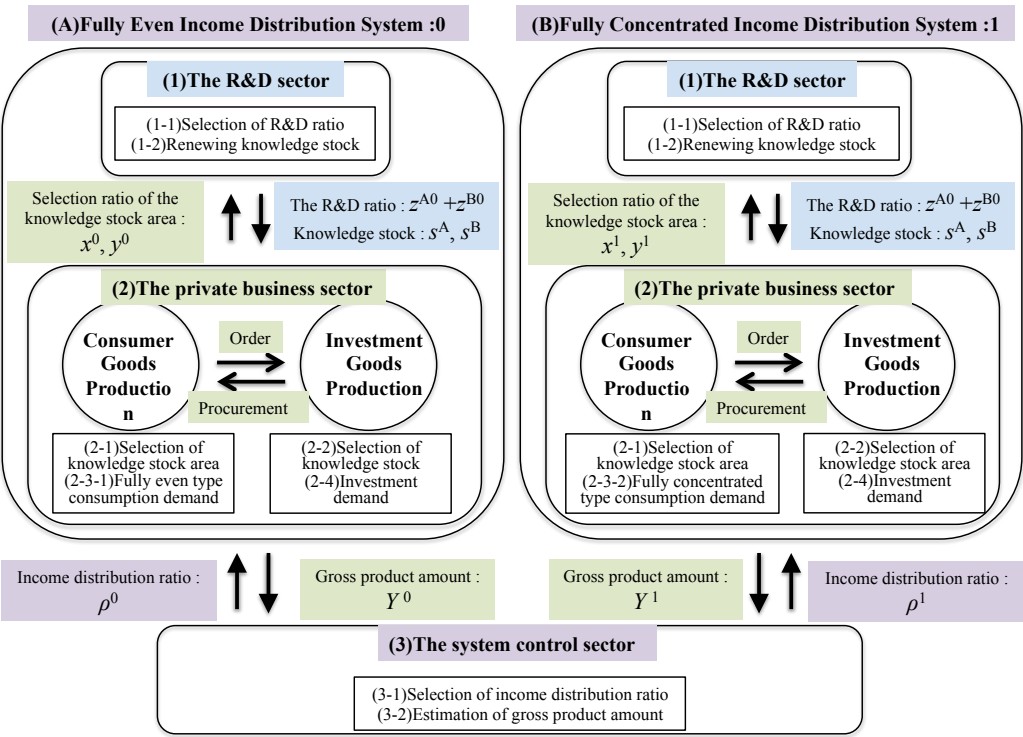

**Figure 3.** The overall structure of an agent-based macroeconomic model in this paper.

Moreover, the decision-making of each sector adopts an adaptive mechanism consistent with the PDCA cycle actually carried out in management, and the macrostate of decision-making of each sector emerges as the demography by replicator dynamics [50,53]. Besides, as the income distribution system, we assume the completely even virtual macroeconomy and the completely concentrated virtual macroeconomy. For this hypothetical economy, we construct an agent-based model based on inter-system control in which the policy authority (system control sector) adaptively determines the allocation ratio between the two based on the gross product amounts (GDPs) while sustaining the economic growth of the real economic system (national economy).

The procedure for constructing an agent-based model and the procedure for carrying out the simulation are shown below in a step-by-step manner in line with Figure 3. Table 1 summarizes the following procedures.

First, in Section 3, the model in this paper defines three sectors of agents responsible for economic activities and policy guidance and defines the state of decision-making in each sector and economic variables related to each sector. In Section 3.1, the R & D division in (1) in Figure 3 determines the selection ratio and R & D ratio for the renewal of two knowledge stock areas that are sources of technological innovation.

In the private business sector in (2) in Figure 3, in Section 3.2, the production of consumer goods is carried out based on the demand for consumer goods (2-1), while the production of investment goods is carried out based on the demand for investment goods (2-2). While comparing and examining the respective levels, the private business sector selects the ratio of knowledge stock areas that generate each demand and uses each knowledge stock as capital equipment. Here, as the structure of the agent-based model in this paper, the economic system related to private economic activities consisting of the above sector is constructed. We assume that a virtual perfectly even income distribution system (A) and perfectly concentrated income distribution system (B) are operated respectively in

this economic system. However, all structures within the economic system, such as decision-making and business relationships, are consistent with the definitions given above, with the exception of the definition of total consumer demand in Section 4.

**Table 1.** Phased workflow procedures in this paper.

| Chapters and Sections of Text | Numbers in Figure 3 | Procedures to Conduct Modeling and Simulation |
|---|---|---|
| Section 3 | | Definition of sectors and economic variables in this model. |
| 3.1 | (1), (1-1) | R & D Sector: Determine R & D ratio. |
| 3.2 | (2) | Private Business Sector: Selection of knowledge stock areas. |
| 3.2 | (2-1) | Consumer Goods Production Division: Selection on the basis of consumption demand. |
| 3.2 | (2-2) | Investment Goods Production Division: Selection on the basis of investment demand. |
| 3.3 | (3-1) | Inter-System Control Sector: Selection of the income distribution system. |
| Section 4 | | Definition of gains as a decision criterion in each sector |
| 4.1 | (1) | R & D Sector: Definition of the relationship between productivity and scarcity of knowledge stock. |
| 4.2 | (1-2) | Specific definitions of each knowledge stock balance after R & D update. |
| 4.3 | (2) | Specific definitions of consumer and investment demand |
| 4.3.1 | (2) | Induction of demand for consumption and investment common to the two virtual income distribution systems. |
| 4.3.2 | (2-3-1) | Definition of consumption demand in a perfectly even income distribution system |
| 4.3.2 | (2-3-2) | Definition of consumption demand in a perfectly concentrated income distribution system |
| 4.3.2 | (2-4) | Definition of investment demand in both systems |
| 4.3.3 | (2) | Examples of differences in levels between the two systems for consumer demand and investment demand |
| 4.4 | (3-2) | Inter-System Control Sector: Specific definition of gross product amounts under each virtual system and real system |
| Section 5 | | Procedures and results of simulations. |
| 5.1 | | Setting of structural parameters and initial values for the model. |
| 5.2 | | Simulations for three stylized cases. |
| 5.3 | | Simulations for OECD nations. |
| Section 6 | | Discussions on appropriateness of income distribution policy recommendation. |

The system control sector, which is the policy authority in (3) of Figure 3, defines a virtual perfectly even income distribution system (A) and perfectly concentrated income distribution system (B) in Section 3.3. The sector then compares and examines the two systems on the basis of the total output produced under each system and determines the ratio that combines the two virtual systems into the actual macroeconomy, that is, the macroeconomic income distribution ratio (3-1). The macroeconomic model in this paper is a poly-agent-based simulation model with a transaction structure consisting of the above three sectors and with a multi-layered hierarchical structure [49].

Next, Section 4 specifically defines the gain for each sector, which is the decision criterion for selecting state variables in each sector defined in Section 3. In the R & D sector in (1) in Figure 3, Section 4.1 first provides a definition of the relationship between productivity and the scarcity of knowledge stock that is consistent with sustainable technological innovation. In Section 4.2, the balance

of each knowledge stock updated through R & D activities is specifically defined based on this selection ratio (1-2).

The private sector in (2) in Figure 3 specifically defines the structure of consumption and investment demand in Section 4.3. Of these, Section 4.3.1 derives both demands from the definition of output for each knowledge stock area common to both distribution systems, Section 4.3.2 defines consumption demand attributable to each system, and Section 4.3.3 illustrates the difference between the two systems in terms of consumption demand and investment demand. Here, in (A), a perfectly even income distribution system (2-3-1), and (B), a perfectly concentrated income distribution system (2-3-2), total consumption demand has the only structurally different definition. On the other hand, demand for investment goods is defined identically in both systems (2-4).

Section 4.4 of the system control sector in (3) in Figure 3 specifically defines the real and each virtual macroeconomic gross product amount (3-2) based on the selection ratio and production value for each knowledge stock produced by the private business sector (2) under both systems (A) and (B).

The following Section 5 describes the procedures and results for performing simulations on the agent-based model defined in Sections 3 and 4. Section 5.1 sets the model's structural parameters and initial values.

In Section 5.2, we set a stylized group consisting of a high (0.91), medium (0.5), and low (0.091) initial marginal propensity to consume as a benchmark, with a population size of $10^8$ = 100 million, in order to confirm and examine the general trends in our model. We set the initial values of the Gini coefficient in increments of 0.01 from 0.01 to 0.99, and carried out calculations corresponding to 3000 round economic transactions to examine the growth potential of the gross product (GDP) and the degree of fluctuation of the Gini coefficient.

Based on these results, Section 5.3 presents economic indicators related to the marginal propensity to consume (estimated value), population, Gini coefficient, and average economic growth rate for selected OECD countries, and then uses this simulation model to calculate the growth potential of the gross product (GDP). For seven of these countries, the initial Gini coefficient values are expanded in increments of 0.01 from 0.1 to 0.9, including actual measurements, to examine the growth potential as an alternative to income distribution policies.

Finally, based on the simulation results described above, Section 6 examines the pros and cons of recommendations on income distribution policy in line with the issues required for agent-based simulation.

## 3. Method of Building a Simulation Model as an Economic System 1: The Decision-Making in the Three Sectors

The basic structure of the model in this paper incorporates the process behind "Macroscopically observed S-shaped growth pathways" as pointed out by Yoshikawa [62], in which the gradually increasing and decreasing returns produced by each successive new knowledge stock change over time. First, the basic structure of this model is configured by alternately arranging two knowledge stock areas A and B over time, which are the sources of technology [58]. Second, we adopt an agent-based model of inter-system control that consists of two virtual economic systems with different income distributions and that manages the ratios between the two systems over time. The decision-making method of each sector in this model is as follows.

### 3.1. Decision-Making in the R & D Sector

The first is the R & D sector, which "updates" the knowledge stock that defines the state of the art. The sector makes decisions on the ratio of the amount of newly generated knowledge to the amount of accumulated knowledge stock, i.e., the ratio of R & D.

Therefore, let us construct a system consisting of a group of agents who act with three alternatives of "update to knowledge stock area A", "update to knowledge stock area B", and "maintain the existing level without renewing any of them" using the existing total knowledge stock balance as a

source. Assuming that the total balance of knowledge stock existing in the entire society during the "$t$" period is "$s_t$", this can be obtained by averaging both the knowledge stock balance existing in the knowledge stock areas A and B. In addition, the stock balances obtained when updating to each knowledge stock area are set to "$rm^A_t$" and "$rm^B_t$", respectively. Also, if the population ratio of the agents who select the first and second alternative is "$z^A_t$" and "$z^B_t$", respectively, the population ratio of the agents who select the third alternative can be expressed as "$1 - z^A_t - z^B_t$". At this time, if the agents evaluate the knowledge stock balance in each area A and B as each gain, the state transitions relating to the population ratios of the agents that select each alternative are defined in the following discrete replicator dynamics [57,63]:

$$z^A_t = \frac{(z^A_{t-1})^2 rm^A_t}{(z^A_{t-1})^2 rm^A_t + (z^B_{t-1})^2 rm^B_t + (1 - z^A_{t-1} - z^B_{t-1})^2 s_{t-1}} \tag{1}$$

$$z^B_t = \frac{(z^B_{t-1})^2 rm^B_t}{(z^A_{t-1})^2 rm^A_t + (z^B_{t-1})^2 rm^B_t + (1 - z^A_{t-1} - z^B_{t-1})^2 s_{t-1}} \tag{2}$$

Next, let us consider the relationship between the population ratio of the agents who choose each alternative to renew their knowledge stock and the average R & D ratio in the entire agent population. For example, consider the population ratio of agents in the system that attempt to substitute and update the existing total knowledge stock balance into knowledge stock area A. This population ratio corresponds to an average ratio at which a representative agent belonging to the system selects the knowledge stock area A when the agent invests the total knowledge stock accumulated so far as the source. In other words, it can be interpreted that the R & D ratio into the knowledge stock area A to the total value generated by the total knowledge stock is obtained (Figure 4).

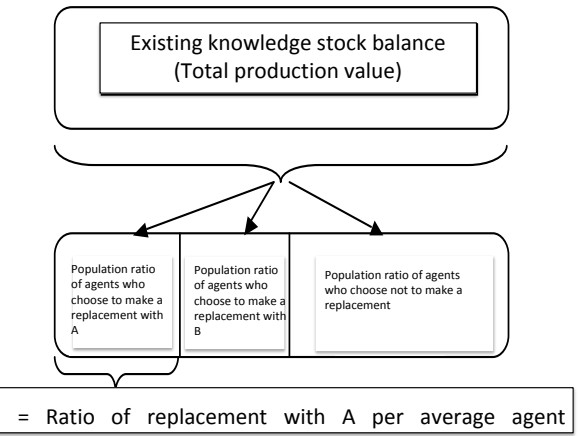

**Figure 4.** Relationship between knowledge stock selection rates derived from replicator dynamics and R & D ratio.

## 3.2. Decision-Making in the Private Business Sector

The second is the private business sector, which consists of two divisions: The consumer goods production division and the investment goods production division. The private business sector makes decisions over time to accumulate more technologically advanced, or more productive, knowledge stock areas A or B. The basis for evaluating the productivity of each knowledge stock is the amount of demand for consumption and investment relative to the production value produced by each knowledge stock. While the consumer goods production division uses the consumption demand amount as the value evaluation standard while the investment goods production division uses the equipment investment demand amount as the value evaluation standard.

First, in the consumer goods production division, if the same knowledge stock area as that of the investment goods production division is selected, capital investment with a higher level of technology, reflecting the results of the R & D sector, can be implemented. As a result, consumer goods can be produced based on the updated new knowledge stock, and the consumer goods production division can obtain the new gains "$u_t$" equal to the amount of demand for consumption. At this time, the investment goods production division can obtain gains "$v_t$" equal to the amount of demand for capital investment reflecting the results of the R & D sector.

On the other hand, if the knowledge stock of a different field is selected in both of the divisions, capital investment reflecting the results of the R & D sector cannot be implemented, and neither the consumer goods production division nor the investment goods production division can implement capital investment and carry out production based on the updated new knowledge stock of the current year. At this time, let us assume that consumption demand "$u_{t-1}$" corresponding to the level of knowledge stock accumulated up to the previous fiscal year occurs in the consumer goods production division while new capital investment demand does not occur in the investment goods production division.

Under the above configuration, we can configure the population dynamics of this sector in which agents in each division select the alternative knowledge stock areas A and B while evaluating and learning the magnitude of the gain obtained based on different value criteria. Assuming that the ratios for selecting the knowledge stock area A in the current period in the consumer goods production division and the investment goods production division are "$x_t$" and "$y_t$", respectively, replicator dynamics, which derives the state transition of the knowledge stock selection in each division, can be expressed as follows [57,63]:

$$x_t = \frac{x_{t-1}y_{t-1}u_t^A + x_{t-1}(1-y_{t-1})u_{t-1}^A}{x_{t-1}y_{t-1}u_t^A + x_{t-1}(1-y_{t-1})u_{t-1}^A + (1-x_{t-1})y_{t-1}u_{t-1}^B + (1-x_{t-1})(1-y_{t-1})u_t^B} \tag{3}$$

$$y_t = \frac{y_{t-1}x_{t-1}v_t^A}{y_{t-1}x_{t-1}v_t^A + (1-y_{t-1})(1-x_{t-1})v_t^B} \tag{4}$$

### 3.3. Decision-Making in the Inter-System Control Sector

The third is the inter-system control sector, which controls the policy goal of "economic growth" and makes decisions on the income distribution ratio of the whole economic system in view of long-term economic growth. The valuation standard for the sector is the value equivalent to the gross product amounts (equivalents of GDP) by the second private business sector. Then, the private business sector is structured as a virtual system of two bipolar institutional requirements.

The institutional requirement relates to the distribution of income. One virtual income distribution system is (1) a system that distributes the national income evenly among all agents that make up the society (perfectly even distribution system ratio: $\rho_0 = 1$) while the other system is (2) a system that concentrates the income on the only one agent that makes up society (perfectly concentrated distribution system ratio: $\rho_1 = 1$). In the inter-system control sector, the ratio between each system ($\rho_0 + \rho_1 = 1$), that is, the income distribution ratio, is calculated over time while the gross product amounts produced under each income distribution system at the two poles are evaluated in comparison with each other (Figure 5).

In the inter-system control sector, first, two kinds of virtual economic systems described later select the allocation ratio of the knowledge stock areas A and B based on each criterion. In each virtual economic system, the total production equivalent amount "$Y^0{}_t$" and "$Y^1{}_t$" generated according to this selection ratio between knowledge stock areas are compared at each period. As a result, the inter-system control sector adaptively selects a more advantageous system among the two virtual economic systems each period and induces a combination ratio of these two, that is, an income distribution ratio as a society as a whole (Figure 6). At this time, in each virtual economic system, the

organizational structure of the R & D sector and the private business sector that produces consumer and investment goods, as well as the economic structure in evaluating and measuring the gross product amounts, are all the same. Only the knowledge stock areas A and B that are adaptively selected under each system, i.e., the difference in the level of technology, define the difference in the gross product amounts of the two systems. We should note that the gross product amount reflecting this level of technology is a valuation criterion for selecting each virtual economic system (see Section 4.4).

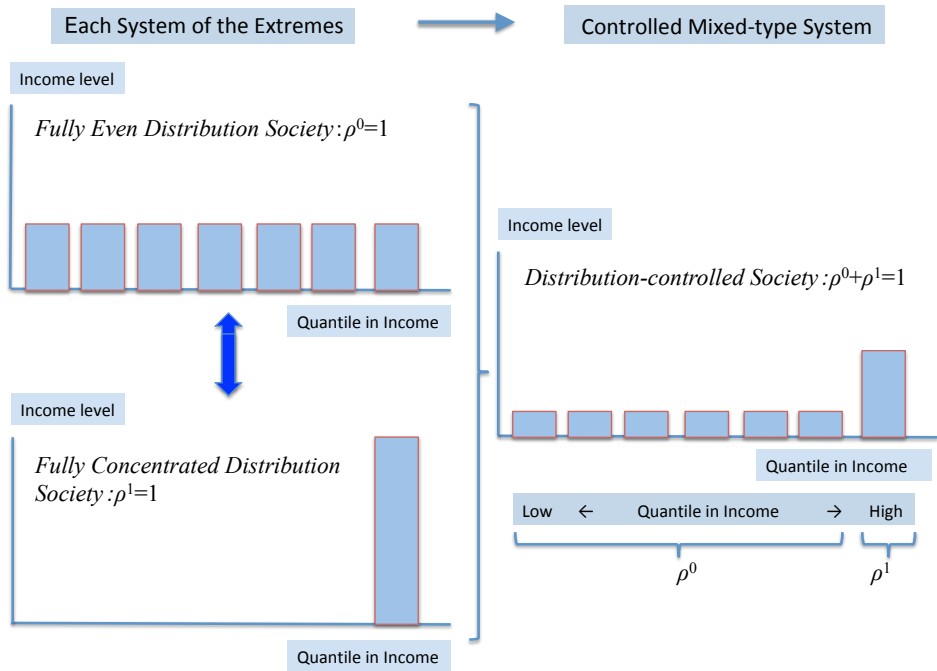

**Figure 5.** Income distribution structure under inter-system control.

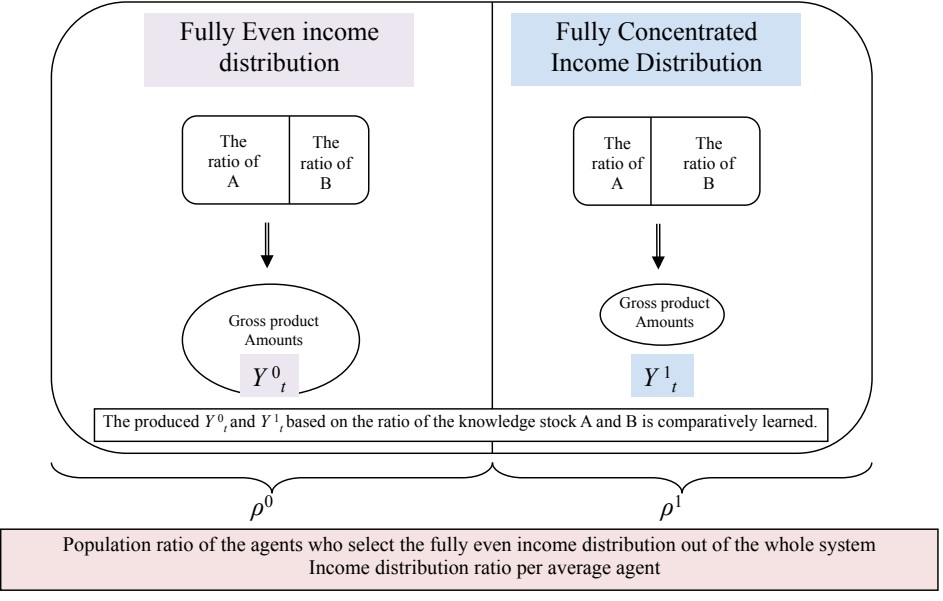

**Figure 6.** Controlling income distribution.

In this sector, each agent exchanges information on the gross product amounts produced under each income distribution system and evaluates them. Replicator dynamics, which derives the state

variable $\rho_t$ for the income distribution system, can be expressed as follows [57,63], based on the population ratio of the agents that have chosen the perfectly even distribution system. Here, $\rho_t = 1$ in the perfectly even distribution system and $\rho_t = 0$ in the perfectly concentrated distribution system. However, when the income distribution system chosen by each agent is different, the level of each gross product amounts realized in the previous term is used as the value evaluation:

$$\rho_t = \frac{\rho_{t-1}^2 Y_t^0 + \rho_{t-1}(1 - \rho_{t-1})Y_{t-1}^0}{\rho_{t-1}^2 Y_t^0 + \rho_{t-1}(1 - \rho_{t-1})Y_{t-1}^0 + (1 - \rho_{t-1})\rho_{t-1}Y_{t-1}^1 + (1 - \rho_t)^2 Y_t^1} \tag{5}$$

## 4. Method of Building a Simulation Model as an Economic System 2: The Internal Organizational Structure and the Related Assumptions in the Three Sectors

This chapter describes the internal organizational structure in the three sectors described in Section 3, and the generation process and the assumptions concerning the gain that serves as the basis for decision-making in each sector.

### 4.1. Assumptions about Scarcity and Productivity of Knowledge Stock

In this paper, we implement the process of evolutionary technological innovation, that is, the process of "new occurrence" and "diffusion" of technology as follows. Suppose, for example, that a highly productive technology has occurred. As the technology becomes more widespread, it becomes obsolete, its productivity declines, and the market in which the technology has been put into practice saturates. At this time, there is a possibility that a new technology with high productivity due to its scarcity, which can be put into practice by only a small number of agents, will be developed. The process of spreading this highly productive technology to society through imitation is implemented in replicator dynamics. In other words, the mechanism involved in sustainable technological innovation, which constitutes the model in this paper, is a combination of two components: One is the implementation of the "genesis process" of technology in the "fluctuations in productivity" defined below, and the other is the implementation of the "diffusion process" of technology in the "replicator dynamics".

We can implement the alternation mechanism between the "occurrence" and "saturation" of the technology as follows. That is, if the scarcity of a technology is expressed based on the population ratio of the agent who selects it, and if the high or low productivity of the technology is normalized to the range of [0,1] based on its scarcity, then we can implement a transition mechanism between the occurrence and saturation of the technology using the transition state [0,1] of the population ratio linked to high or low productivity.

In this paper, two types of knowledge stock areas A and B are supposed to represent the sources of these technologies. Thus, we can implement the mechanism of increasing and decreasing returns, which exists behind the "macroscopically observed S-shaped growth pathways" as pointed out by Yoshikawa [62], and is produced by each new knowledge stock over time.

Therefore, this model assumes that the scarcity of knowledge stock varies depending on the selection of knowledge stock areas A and B in the "private business sector", and thus the productivity derived from each knowledge stock fluctuates. For example, in the production activities in the "consumer goods production division" and the "investment goods production division", if the population ratio "$x_{t-1}$" and "$y_{t-1}$", which select knowledge stock area A at the end of the previous period (the beginning of the current period), rise together, it is considered that the scarcity of the knowledge stock area A decreases and its productivity decreases (Equation (6)). On the other hand, the scarcity of the knowledge stock area B is defined using the population ratio "$1 - x_{t-1}$" and "$1 - y_{t-1}$" of which each sector selects (Equation (7)).

In this paper, we define the new amount of knowledge generated and accumulated each year in each knowledge stock area using the following relational equation based on the range of [0, 1] (see Equations (6) and (7)). The "$c$" is a parameter for controlling the degree of concentration of marginal productivity in the selection of the knowledge stock area:

$$\left(1 - \frac{x_{t-1} + y_{t-1}}{2}\right)^c; \, c \geq 1 \tag{6}$$

$$\left(1 - \frac{(1 - x_{t-1}) + (1 - y_{t-1})}{2}\right)^c; \, c \geq 1 \tag{7}$$

*4.2. The Internal Structure in the R & D Sector: The Generation Process of Knowledge Stock and Assumptions about Gain Structure*

This section describes requirements that are assumed for the process of generating R & D results that are the criterion for evaluating decisions in this sector, as an internal structure that defines R & D activities. The gain structure, which is the criterion for evaluating decisions, is defined as follows. First, let's consider the case where the average knowledge stock balance per capita in the entire system, "$s_{t-1}$", which has been accumulated up to the previous period, is all invested to generate a new knowledge stock area A. In this case, the source of funds for R & D investment shall be "$s_{t-1}$" itself. The entire system here is a real economic system that integrates a virtual, perfectly even distribution system with a perfectly concentrated distribution system.

Consider the following relationship in which the return on R & D investment depends on the proportion of the R & D population that employs knowledge stock area A. In other words, the smaller the population ratio "$z^A_{t-1}$" as of the end of the previous period (the beginning of the current period), the higher the scarcity of the knowledge stock area A; on the other hand, the larger the population ratio, the more the commercial value of the area becomes obsolete and the lower the profit. Further, we assume that this result is subject to the scarcity of each knowledge stock area A and B selected by the private business sector as represented by Equations (6) and (7).

By formulating the above relationship, the gains "$rm^A_t$" and "$rm^B_t$" are expressed as the profits when the average knowledge stock balance up to the end of the previous period is updated into the respective knowledge stock areas, as follows (Equations (8) and (9)). We should note that the parameter "$a$" affects the magnitude of R & D revenue, and as a result, it has a function of controlling the degree to which high marginal productivity concentrates in knowledge stock areas with a low selection ratio, together with the parameter "$c$". In this paper, we construct a skewed distribution in the sense of Scherer [64] by setting $a = 1.8$ and $c = 9.4$, where large amounts of remuneration concentrate to some extent on some agents:

$$rm^A_t = s_{t-1}\left(\left(\frac{1}{z^A_{t-1}}\right)^a - 1\right)\left(1 - \frac{x_{t-1} + y_{t-1}}{2}\right)^c; \, a \geq 1, c \geq 1 \tag{8}$$

$$rm^B_t = s_{t-1}\left(\left(\frac{1}{z^B_{t-1}}\right)^a - 1\right)\left(1 - \frac{(1 - x_{t-1}) + (1 - y_{t-1})}{2}\right)^c; \, a \geq 1, c \geq 1 \tag{9}$$

The population ratio of agents renewing in knowledge stock areas A and B at the beginning of the current period is "$z^A_{t-1}$" and "$z^B_{t-1}$", respectively, and the population ratio of agents not renewing is "$1 - z^A_{t-1} - z^B_{t-1}$". Therefore, the average balance "$s^A_t$" and "$s^B_t$" of each knowledge stock in the current period after renewal are expressed by averaging through each population ratio as follows (Equations (10) and (11)). The average balance "$s_t$" of the entire system is configured as follows (Equation (12)) by averaging through the selected ratio of the knowledge stock areas A and B by the agents belonging to the system:

$$s^A_t = \left(1 - z^A_{t-1} - z^B_{t-1}\right)s^A_{t-1} + z^A_{t-1}\, rm^A_t; \, t = 1, 2, 3, \cdots \tag{10}$$

$$s^B_t = \left(1 - z^A_{t-1} - z^B_{t-1}\right)s^B_{t-1} + z^B_{t-1}\, rm^B_t; \, t = 1, 2, 3, \cdots \tag{11}$$

$$s_t = z_{t-1} s_t^A + (1 - z_{t-1}) s_t^B; \ z_{t-1} = \frac{z_{t-1}^A}{z_{t-1}^A + z_{t-1}^B} \tag{12}$$

*4.3. The Internal Structure in the Private Business Sector: Diminishing Marginal Propensity to Consume and Income Distribution*

The private business sector, which is controlled by the inter-system control sector, is under the bipolar income distribution system introduced in the previous chapter, which consists of two virtual systems: (1) A perfectly even distribution system and (2) a perfectly concentrated distribution system. This section describes the internal structure that defines each virtual system and the requirements that are assumptions for each of the consumption and investment demand that serve as the basis for evaluating decisions in the sector.

4.3.1. The Generation Process of Consumption Demand and Investment Demand

In this model, under the economic system with two bipolar income distributions, the consumer goods production division and the investment goods production division select knowledge stock areas A and B each period. The consumer goods production division, which adopts "level of consumption demand" as the evaluation criterion, procures investment goods from the investment goods production division, which adopts "level of investment demand" as the evaluation criterion, and then makes capital investment related to the knowledge stock area selected by itself, and then conducts production.

Let us now express the marginal products produced in each knowledge stock area during the fiscal year under review as "$m_t^A$" and "$m_t^B$". This marginal product is calculated by multiplying the average balance of knowledge stock "$s_t$" of the entire system updated by the R & D Sector by the R & D ratio "$z_{t-1}^A + z_{t-1}^B$" as of the beginning of the current period (end of the previous term), and we assume that the scarcity of each knowledge stock area is subject to Equations (6) and (7) (Equations (13) and (14)):

$$m_t^A = (z_{t-1}^A + z_{t-1}^B) s_t \left( 1 - \frac{x_{t-1} + y_{t-1}}{2} \right)^c; \ c \geq 1 \tag{13}$$

$$m_t^B = (z_{t-1}^A + z_{t-1}^B) s_t \left( 1 - \frac{(1 - x_{t-1}) + (1 - y_{t-1})}{2} \right)^c; \ c \geq 1 \tag{14}$$

At this time, the amount of production "$Y_t^A$" and "$Y_t^B$" generated by each knowledge stock area shall be constituted by subtracting the amount equivalent to R & D expenses for both knowledge stock areas from the capital stock balance "$sk_{t-1}^A$" and "$sk_{t-1}^B$" related to each existing knowledge stock area, and replacing them with marginal products that are the product of R & D activity in the current period (Equations (15) and (16)). This total output is equal to the knowledge stock balance in each area. This is equivalent to the AK type production function defined as A = 1:

$$Y_t^A = \left( 1 - z_{t-1}^A - z_{t-1}^B \right) sk_{t-1}^A + m_t^A; \ t = 1, 2, 3, \cdots \tag{15}$$

$$Y_t^B = \left( 1 - z_{t-1}^A - z_{t-1}^B \right) sk_{t-1}^B + m_t^B; \ t = 1, 2, 3, \cdots \tag{16}$$

Next, let us define the valuation gain "$u_t$" of the consumer goods production division. We assume that the valuation gain of the consumer goods production division is equal to the consumption level "$C_t$" realized at the end of the current term, which is an increasing function of the gross product amount, "$Y_t$", and the marginal propensity to consume diminishes (see Appendix A). Therefore, in this model, the qualitative nature of the diminishing marginal propensity to consume is expressed by natural logarithms as follows. Gain evaluations related to the knowledge stock areas A and B are expressed by Equations (17) and (18), respectively. The reason why the argument of the logarithm is designated as "$Y_t + 1$" is that " $C_t$" is translated to a positive value for $Y_t > 0$ and normalized. In discrete replicator dynamics, the gain evaluations must be positive:

$$u_t^A = C_t^A = \log\!\left(Y_t^A + 1\right) \tag{17}$$

$$u_t^B = C_t^B = \log\!\left(Y_t^B + 1\right) \tag{18}$$

As a result, the marginal propensity to consume is positive and 1 or less for $Y_t > 0$:

$$\frac{dC_t}{dY_t} = \frac{1}{Y_t + 1} \leq \text{ for } \forall Y_t \geq 0$$

On the other hand, the profit evaluation "$v_t$" that serves as the code of conduct for the investment goods production division is the investment demand level "$I_t$" that is funded by funds loaned from the current savings amount "$S_t$". In this paper, assuming that the investment demand is proportional to the loan amount from the savings amount "$S_t$" and is subject to the scarcity of each knowledge stock area (Equations (6) and (7)), the demand is defined as Equations (19) and (20) as follows:

$$v_t^A = I_t^A = S_t^A \times \left(1 - \frac{x_{t-1} + y_{t-1}}{2}\right)^{c}; \ S_t^A = Y_t^A - C_t^A \tag{19}$$

$$v_t^B = I_t^B = S_t^B \times \left(1 - \frac{(1 - x_{t-1}) + (1 - y_{t-1})}{2}\right)^{c}; \ S_t^B = Y_t^B - C_t^B \tag{20}$$

### 4.3.2. Income Distribution System and Consumer Demand

Based on these definitions, we can define the evaluation standard of gains in two virtual bipolar income distribution systems. Therefore, the total population of the society that constitutes this system is defined as "$n$". First, in the perfectly even distribution type, since the gross product amount (income amount) is distributed equally among all "$n$" members of society, the amount of consumption per capita occurs depending on the level of income per capita, and the total amount is an aggregated value as the gross consumption expenditure of society. As a result, in the society of this system, the gains defined based on the total consumption demand and the total investment demand derived from each knowledge stock area can be expressed as follows. Here, each variable of the perfectly even distribution type is represented by adding the subscript "0" to the right shoulder (Equations (21)–(24)):

$$u_t^{A0} = n \log\!\left(\frac{Y_t^{A0}}{n} + 1\right) \tag{21}$$

$$v_t^{A0} = S_t^{A0} \times \left(1 - \frac{x_{t-1}^0 + y_{t-1}^0}{2}\right)^{c} \tag{22}$$

$$u_t^{B0} = n \log\!\left(\frac{Y_t^{B0}}{n} + 1\right) \tag{23}$$

$$v_t^{B0} = S_t^{B0} \times \left(1 - \frac{(1 - x_{t-1}^0) + (1 - y_{t-1}^0)}{2}\right)^{c} \tag{24}$$

On the other hand, in the perfectly concentrated type, the gross product amount (income amount) is distributed to only one agent in the society, and the income distribution of the remaining $n - 1$ agents is defined as 0. At this time, the gain amount defined from the total consumption demand and the total investment demand of the society constituting the system can be expressed as follows, respectively, for this single agent. Here, each variable in the case of the perfectly concentrated distribution type is represented by appending 1 to the right shoulder (Equations (25)–(28)):

$$u_t^{A1} = \log\!\left(Y_t^{A1} + 1\right) \tag{25}$$

$$v_t^{A1} = S_t^{A1} \times \left(1 - \frac{x_{t-1}^1 + y_{t-1}^1}{2}\right)^c \tag{26}$$

$$u_t^{B1} = \log\left(Y_t^{B1} + 1\right) \tag{27}$$

$$v_t^{B1} = S_t^{B1} \times \left(1 - \frac{(1 - x_{t-1}^1) + (1 - y_{t-1}^1)}{2}\right)^c \tag{28}$$

Two virtual bipolar income distribution systems, one with a perfectly even distribution and the other with a perfectly concentrated distribution, are integrated into the real economic system by the inter-system control sector. The difference between the two systems, and therefore the object for the integration, is only the total consumption demand. That is, according to the selection ratio "$\rho_t$" of the virtual income distribution system controlled by this sector at the beginning of the current period, the linear combination of the total consumption demand (Equations (21) and (23), and Equations (25) and (27)) derived from each knowledge stock area constitutes the gain evaluation standard for the consumer goods production division in the real economic system (Equations (29) and (30)):

$$u_t^A = \rho_{t-1}u_t^{A0} + (1 - \rho_{t-1})u_t^{A1} \tag{29}$$

$$u_t^B = \rho_{t-1}u_t^{B0} + (1 - \rho_{t-1})u_t^{B1} \tag{30}$$

### 4.3.3. Consumption Structure with a Diminishing Marginal Propensity to Consume

The assumption about the consumption function adopted in this paper is that "The marginal propensity to consume diminishes." At this time, as illustrated in the logarithmic formulation above, in the aggregate consumption demand criterion, the perfectly even distribution type is superior to the perfectly concentrated distribution type in all the domains (Figure 7a). At any level of gross income, the marginal propensity to consume of income distributed per capita is relatively higher than that of the gross income level because Equations (21) and (23) divide gross income equally among the total population. Therefore, at all income levels, the total amount of consumption, which is the sum of the per capita consumption demand, exceeds the total amount of consumption in Equations (25) and (27), where gross income is under the monopoly of only one agent. On the other hand, in the investment demand standard, the perfectly concentrated distribution type is superior (Figure 7b).

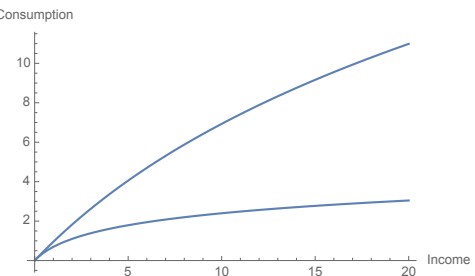

**(a)** Consumer demand curve. Note: The upper curve with perfectly even income distribution type, lower curve with perfectly concentrated income distribution type.

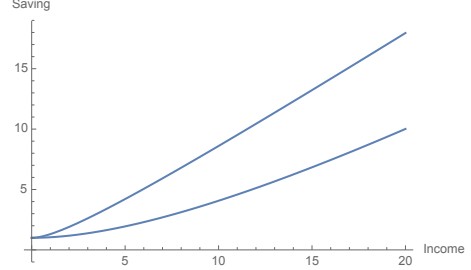

**(b)** Saving curve. Note: The upper curve with perfectly concentrated income distribution type, the lower curve with perfectly even income distribution type.

**Figure 7.** The diagrams show consumer demand (left) and saving (right) due to differences in the forms of the income distribution.

We should note that in this model, the structures that generate demand for consumption and investment do not directly affect the selection of the virtual income distribution systems. When consumption and investment are carried out under the different income distribution systems mentioned

above, system selection is under control by evaluating the total output generated by both systems through the process of selecting knowledge stock areas A and B with higher productivity. In other words, it is a system that evaluates and controls the performance of sustainable growth, that is, and that selects a knowledge stock area with higher productivity over time, in a perfectly even distribution system in which consumption demand occupies a relatively large weight, and in a perfectly concentrated distribution system in which investment demand is leading. We should note that the model presented in this paper is not a simple one in which the formulation of the above model itself directly affects system selection, but rather provides income distribution system selection that is consistent with the path of sustained technological innovation.

We will introduce the production process characteristics of each virtual system derived from these assumptions as below. First, in a perfectly even income distribution system, the weight of consumption demand is relatively large. As previously assumed, consumption demand, which is the criterion for evaluating the gain of the consumer goods production division, is in a monotonously increasing function concerning income level and is less susceptible to marginal productivity fluctuations in the knowledge stock area as income levels rise. As a result, we can say that a perfectly even income distribution system is one that defines stable economic growth in which stable consumption demand relative to income leads to the selection of knowledge stock areas.

On the other hand, in a perfectly concentrated type, the weight of investment demand is relatively large. In the investment goods production division, the gain evaluation criteria are assumed to depend on investment demand and scarcity of knowledge stock areas, so the model structure is sensitive to changes in marginal productivity. As a result, we can say that under the perfectly concentrated distribution system, the demand for investment, which responds flexibly to changes in marginal productivity, leads to the selection of knowledge stock areas, and that this is a system that defines technological change.

*4.4. Internal Structure in the Inter-System Control Sector: the Generation Process of Gross Product Equivalent*

The inter-system control sector calculates the total output of an economic system consisting of the R & D and private business sector for each of two virtual income distribution system requirements: A perfectly even system and a perfectly concentrated system. This sector controls the two types of virtual systems through the income distribution ratio, which is the ratio by which each system is selected. This section describes requirements that are assumed for the process of generating gross product equivalent, which is this sector's criterion for evaluating decision-making, as the internal structure that defines this sector.

As with the previously defined Equations (15) and (16), the production amount "$Y^A_t$" and "$Y^B_t$" generated by each knowledge stock area A and B shall consist of the capital stock "$sk^A_{t-1}$" and "$sk^B_{t-1}$" associated with each existing knowledge stock area, respectively, after deducting the R & D expenditure for each knowledge stock area and replacing them with marginal products that are the product of current R & D activities. This production amount equals the capital stock balance in each area. That is, it is defined as A = 1 in the AK-type production function.

The construction procedure of the production amount is the same for the virtual economic systems, that is, perfectly even distributed systems (Right shoulder subscript "0") and perfectly concentrated distributed systems (right shoulder subscript "1"), and the real economic systems (no right shoulder subscript) controlled by the selection ratio of the two virtual systems chosen by this sector (Equations (31), (32), (33), (34), (35), and (36), respectively). It is assumed, however, that the capital stock balance of the controlled real economic system is shared at the beginning of the current period, and that marginal products are accumulated by the selection ratio of the knowledge stock area in the consumer goods production division of each income distribution-type system at the end of the current period:

$$Y_t^{A0} = \left(1 - z_{t-1}^{A0} - z_{t-1}^{B0}\right)sk_{t-1}^A + x_t^0 m_t^{A0}; \; t = 1, 2, 3, \cdots \tag{31}$$

$$Y_t^{B0} = \left(1 - z_{t-1}^{A0} - z_{t-1}^{B0}\right)sk_{t-1}^{B} + \left(1 - x_t^0\right)m_t^{B0}; \; t = 1, 2, 3, \cdots \tag{32}$$

$$Y_t^{A1} = \left(1 - z_{t-1}^{A1} - z_{t-1}^{B1}\right)sk_{t-1}^{A} + x_t^1 m_t^{A1}; \; t = 1, 2, 3, \cdots \tag{33}$$

$$Y_t^{B1} = \left(1 - z_{t-1}^{A1} - z_{t-1}^{B1}\right)sk_{t-1}^{B} + \left(1 - x_t^1\right)m_t^{B1}; \; t = 1, 2, 3, \cdots \tag{34}$$

$$Y_t^{A} = \left(1 - z_{t-1}^{A} - z_{t-1}^{B}\right)sk_{t-1}^{A} + x_t m_t^{A}; \; t = 1, 2, 3, \cdots \tag{35}$$

$$Y_t^{B} = \left(1 - z_{t-1}^{A} - z_{t-1}^{B}\right)sk_{t-1}^{B} + (1 - x_t)m_t^{B}; \; t = 1, 2, 3, \cdots \tag{36}$$

The average (aggregated) of the production amount of each knowledge stock area A and B produced by each of the above systems in accordance with the selection ratio of the knowledge stock area in the consumer goods production division and the investment goods production division at the end of the current period corresponds to the gross product amounts (Equations (37), (38), and (39), respectively). However, when the knowledge stock areas selected in the consumer goods production division and the investment goods production division are different, the gross product amounts at the end of the previous period in the actual economic system are evaluated. This sector selects both virtual systems over time based on the gross product amounts (Equations (37) and (38)), which are produced in the perfectly even distribution system and the perfectly concentrated distribution system, respectively, as a criterion for gain evaluation:

$$Y_t^0 = x_t^0 y_t^0 Y_t^{A0} + x_t^0 (1 - y_t^0) Y_{t-1}^{A} + (1 - x_t^0) y_t^0 Y_{t-1}^{B} + (1 - x_t^0)(1 - y_t^0) Y_t^{B0}, \tag{37}$$

$$Y_t^1 = x_t^1 y_t^1 Y_t^{A1} + x_t^1 (1 - y_t^1) Y_{t-1}^{A} + (1 - x_t^1) y_t^1 Y_{t-1}^{B} + (1 - x_t^1)(1 - y_t^1) Y_t^{B1} \tag{38}$$

$$Y_t = x_t y_t Y_t^{A} + x_t (1 - y_t) Y_{t-1}^{A} + (1 - x_t) y_t Y_{t-1}^{B} + (1 - x_t)(1 - y_t) Y_t^{B} \tag{39}$$

## 5. Results

So far, we have described the two virtual income distribution systems, i.e., the perfectly even distribution system and the perfectly concentrated distribution system, the function of the system that controls between the two systems, and the structure of each sector in this model. Can control between the two systems lead to a more viable economic system through technological innovation, that is, continuous change of knowledge stock? If so, what is the income distribution ratio? In this chapter, we will try to answer these policy issues by simulation. This chapter uses Mathematica 10 to implement the models described in Sections 2–4.

### 5.1. Prerequisites: Setting Parameters and Initial Values for Variables

The variables in this simulation model are set as follows, according to previous papers [57–59]. First, there are structural parameters relating to the reward of R & D activities and the marginal productivity of knowledge stock, set at $a = 1.8$ and $c = 9.4$, respectively. These settings are the conditions under which long-term economic growth can be stably performed through technological change, i.e., the accumulation of knowledge stock over time in this model.

Next, the initial value for the selection ratio of the knowledge stock based on the knowledge stock area A is set at 0.4 in both the consumer goods production and the investment goods production division, and the initial value of the renewal ratio of the knowledge stock (R & D ratio) in the R & D sector is set at 0.01.

Furthermore, for the income distribution ratio, the initial state was performed in increments of 0.01 from 0.01 to 0.99 with respect to the perfectly even distribution system ratio, $\rho_0$.

Gross product amount (gross income) consists of the knowledge stock balance at the end of the previous period, less R & D costs, and the resulting marginal products. Because of the logarithmic consumption function, the marginal propensity to consume diminishes rapidly with increasing income. Therefore, the initial value of the knowledge stock balance and the population size is set at an adequate

value to clarify the impact of consumption demand in the perfectly even distribution system. Therefore, the population is fixed at $10^8$ = 100 million, and the initial knowledge stock balance is set at $10^7$, $10^8$, and $10^9$. The corresponding initial values of the marginal propensity to consume are (1) 0.91 (high), (2) 0.5 (middle), and (3) 0.091 (low), and the following calculation is made for each of the three cases.

In the following simulation, "growth potential", that is, "the gross product amount after trial/initial value of the gross product amount" is calculated for a period up to "3000 times" or "immediately after lock-in". One period represents one cycle of economic activity in each division. For example, while the consumer goods production and investment goods production divisions transact between them by selecting a knowledge stock area, this single decision-making corresponds to one cycle described above. We should note that the cycles associated with such decision-making do not necessarily correspond to one accounting period. Needless to say, it is possible to transact many times in one accounting period.

*5.2. Simulation Results (1): Basic Trends in the Model*

Figure 8 summarizes the results of the growth potential after the implementation of 3000 economic activities with each initial Gini coefficient values ranging from 0.01 to 0.99 in increments of 0.01 for each of the following three cases. In the first case, an economy with a high marginal propensity to consume in the initial state, the growth potential is highest when the initial state of the Gini coefficient is among 0.01 and 0.5 scant. We can describe this economy as a consumption-driven economy in which the growth of aggregate consumption demand is sustained by distributing gross income equally to each citizen (see Figure 7). The growth is thought to be sustainable through consumption-led knowledge stock selection over time. However, as the initial state of the Gini coefficient exceeds 0.6, the growth potential becomes the lowest with large fluctuations of the three cases (Figure 8).

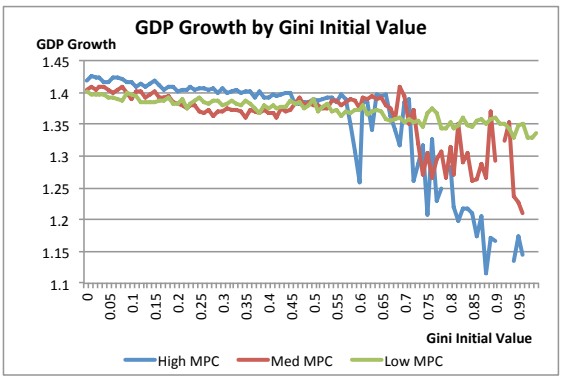

**Figure 8.** The diagram shows the simulation results of the GDP growth potentials for the three cases.

On the other hand, in the third case, an economy with a low marginal propensity to consume at the initial state, the growth potential tends to be relatively low when the initial state of the Gini coefficient at 0.01 and 0.45 scant. We can regard this economy as a mature consumer demand economy in which the growth of total consumption demand cannot be sustained enough even if gross income is distributed evenly per capita. As a result, sufficient growth is considered not to be achieved through consumer-led knowledge stock selection. However, as the initial state of the Gini coefficient increases from 0.01 to 0.99, the growth potential gradually decreases but is maintained at about 1.3 times, and its dispersion unexpectedly becomes much smaller than the other two cases. This shows that even in an economy with a high concentration of income, it is possible to maintain a stable society with investment-led growth and the small dispersion of the growth potential, though the investment demand is configured by the fluctuating selections of each knowledge stock area (see Equations (19) and (20)).

Next, we examined how the Gini coefficient fluctuates in the course of economic growth depending on its initial state. In general, as illustrated in Figure 9, in the range where the initial value is 0.5 or less, the control of the growth path sustainably performs while fluctuating substantially each around the initial value.

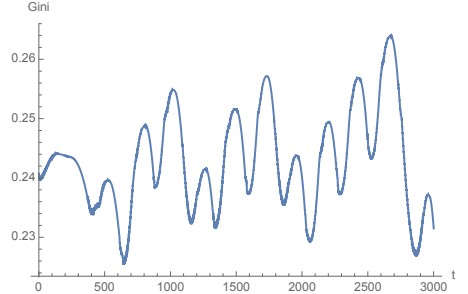

**(a)** MPC is high and Gini initial value = 0.24.

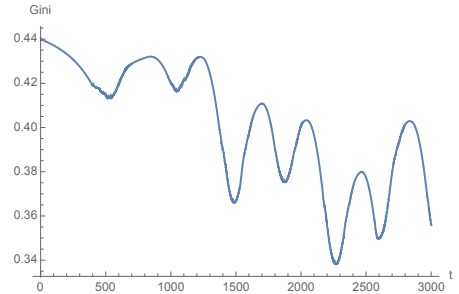

**(b)** MPC is low and Gini initial value = 0.44.

**Figure 9.** The diagram shows the controlled Gini coefficient time series.

Figure 10 summarizes the results of the "standard deviation/mean" of the time series transition of the Gini coefficient after the implementation of 3000 economic activities with each initial Gini coefficient values ranging from 0.01 to 0.99 in increments of 0.01 for each of the following three cases.

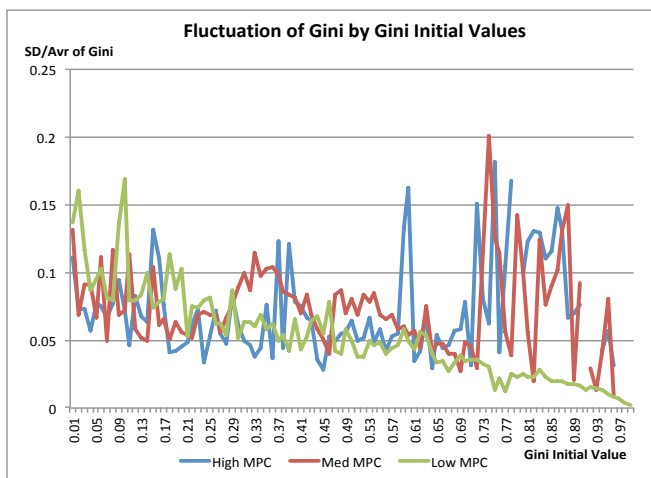

**Figure 10.** The diagram shows the variability of Gini coefficients in the three cases. Note: The vertical axis of the figure is the standard deviation/average of the time series transition of the Gini coefficient. The horizontal axis of the figure is the initial values of the Gini coefficient.

Let us start with the first case. In an economy with a high marginal propensity to consume at the initial stage (consumption-driven economy) and where the initial value of the Gini coefficient is low at less than 0.1, the fluctuation of the Gini coefficient is almost smaller than in other cases, that is, income distribution equality tends to be stably maintained. As equality decreases in the initial state, the change in the Gini coefficient tends to increase and fluctuate at a larger level. According to the simulation results of the time series transition of the Gini coefficient, the Gini coefficient tends to fluctuate with a great decrease.

On the other hand, in the third case, an economy with a low marginal propensity to consume at the initial stage (mature economy in consumption demand), the fluctuation of the Gini coefficient decreases as its initial value increases from 0.01 to 0.99. In other words, in a society where the initial state of the Gini coefficient is low and equality is high, the fluctuation of the Gini coefficient is large, and on the other hand, in a society where equality is low, the fluctuation of the Gini coefficient decreases. As a result of simulation of the time series transition of the Gini coefficient, the alternate change of the knowledge stocks is inactive in the society with low equality, and as a result, the fluctuation of the Gini coefficient becomes small.

From the simulation results of the three stylized cases described above, we can present the following scenarios. In countries and societies with economic structures where the marginal propensity

to consume has increased or is rising due to technological innovation, policies to raise the equality of income distribution by lowering and stably inducing the Gini coefficient are effective for sustainable economic growth.

On the other hand, in a country or society with a mature economic structure where the marginal propensity to consume has declined under the existing technological level, a decline of equality in income distribution has a relatively small impact on economic growth (see Figure 8). Additionally, in this mature economy, there is a positive correlation between economic growth and changes in the Gini coefficient (Figure 11). However, fluctuations in the Gini coefficient tend to be large in economies with a high degree of equality in the initial state (Figure 10). Thus, in the mature economy, policies that, while maintaining overall equality, allow temporary declines of equality and incorporate the results of an investment-led economy are effective.

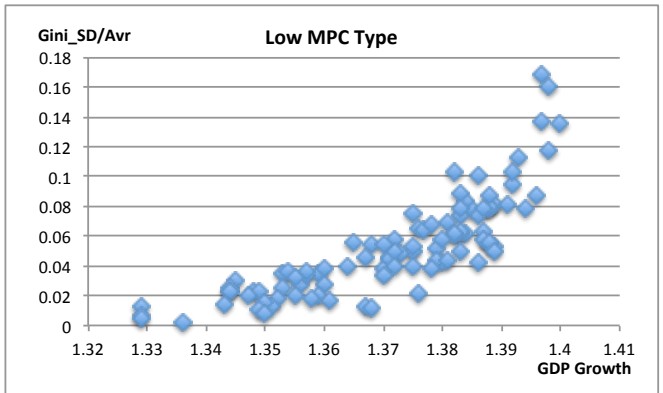

**Figure 11.** The diagram shows the correlation between GDP growth potential and Gini coefficient variability in the mature economies. Note: The vertical axis of the figure is the standard deviation/average of the time series transition of the Gini coefficient. The horizontal axis of the figure is the GDP growth potential.

*5.3. Simulation Results (2): Growth Potential Based on Actual Data of OECD Countries*

To consider the impact of income distribution structures, we relinquish the unique economic structures of each country and use the model structures in this paper in common. Based on the actual data of each country (Table 2), the simulation was carried out with the Gini coefficient as actual data, and the marginal propensity to consume and the knowledge stock balance estimated based on the actual data, as initial values, respectively, while fixing the population as a structural characteristic.

The target countries are the following 17 OECD countries, for which disclosure of the Gini coefficient and estimation of marginal propensity to consume are commonly available. Estimates of marginal propensity to consume from 1970 to 1980s fall into the following categories:

The United Kingdom and the United States, with their high marginal propensity to consume in excess of 0.6;

low-lying countries, centered on northern Europe are 0.4 over to 0.5 under;

and middle-ranking countries are 0.5 over to 0.6 under, including France, Germany, and Switzerland, and in between.

Categories of the countries with high, medium, or low marginal propensity to consume are similar between 1990 and the 2010s except for Japan and Finland. At the same time, the Gini coefficient is also high in the United Kingdom and the United States, where the marginal propensity to consume is high, while it is low in the northern Europe countries.

**Table 2.** Estimates of marginal propensity to consume, and actual Gini coefficient, population, and growth rate for selected 17 OECD countries.

| | MPC | | Gini | Population (1K) | Growth | |
|---|---|---|---|---|---|---|
| | 70s~80s | 90s~10s | 2015 | 2016 | 70s~80s | 90s~10s |
| United Kingdom | 0.646465217 | 0.601914206 | 0.36 | 65,789 | 7.957648279 | 4.126458311 |
| United States | 0.630394714 | 0.677171909 | 0.39 | 322,180 | 8.651048175 | 4.319499858 |
| Belgium | 0.574421456 | 0.47800158 | 0.268 | 11,358 | 8.005920041 | 4.008065682 |
| Switzerland | 0.572077173 | 0.482664205 | 0.297 | 8402 | 6.973479037 | 4.03100397 |
| Germany | 0.570529126 | 0.495904766 | 0.289 | 81,915 | 7.815166096 | 3.767644771 |
| Austria | 0.559876905 | 0.485363264 | 0.276 | 8712 | 8.192717705 | 4.212858548 |
| Australia | 0.556005234 | 0.554664879 | 0.337 | 24,126 | 8.478227701 | 5.271908232 |
| France | 0.555354334 | 0.518755819 | 0.295 | 64,721 | 8.399057604 | 3.848739998 |
| New Zealand | 0.547902397 | 0.559156682 | 0.349 | 4661 | 7.634287798 | 5.023491742 |
| Canada | 0.532391844 | 0.553760096 | 0.318 | 36,290 | 8.856135173 | 4.07049973 |
| Japan | 0.520242224 | 0.610375596 | 0.336 | 126,933 | 9.816553768 | 2.867719949 |
| Luxembourg | 0.492793775 | 0.24632912 | 0.284 | 576 | 9.021792787 | 6.390966269 |
| Denmark | 0.490763018 | 0.440207624 | 0.256 | 5712 | 7.498641034 | 4.408167217 |
| Netherlands | 0.490074326 | 0.406882642 | 0.303 | 16,987 | 7.935791594 | 4.307330483 |
| Finland | 0.487236623 | 0.52907638 | 0.26 | 5503 | 8.949807544 | 3.772866889 |
| Sweden | 0.466007749 | 0.419781897 | 0.278 | 9838 | 7.488521911 | 4.004401052 |
| Norway | 0.460740222 | 0.36105076 | 0.272 | 5255 | 9.070635952 | 5.25636354 |

Note: The growth rate in the table is the average annual growth rate for each period.

Table 3 shows the results of a simulation of growth potential ("the gross product amount after the trial/initial value of the gross product amount") using the estimated marginal propensity to consume and the Gini coefficient as the initial values in the 17 countries. Looking at the relationship between the "marginal propensity to consume/Gini coefficient" and "growth potential", only Luxembourg deviates significantly, but other countries show a positive correlation (Figure 12). In other words, the simulation results show that in the relationship between the marginal propensity to consume and the Gini coefficient, in 16 countries excluding Luxembourg, the higher the marginal propensity to consume (the more obvious the consumption drive), or the more uniform the distribution of income, the higher the growth potential. There is no particular correlation between the initial values of either the marginal propensity to consume or the Gini coefficient and the growth potential.

**Table 3.** Simulation results of growth potential in major OECD countries.

|  | MPC | Population (1K) | Gini | Growth |
|---|---|---|---|---|
|  | 90s~10s | 2016 | 2015 | (Calculated) |
| United Kingdom | 0.601914206 | 65,789 | 0.36 | 1.37651 |
| United States | 0.677171909 | 322,180 | 0.39 | 1.36934 |
| Japan | 0.610375596 | 126,933 | 0.336 | 1.37 |
| Belgium | 0.47800158 | 11,358 | 0.268 | 1.36529 |
| Switzerland | 0.482664205 | 8402 | 0.297 | 1.3678 |
| Germany | 0.495904766 | 81,915 | 0.289 | 1.36972 |
| Austria | 0.485363264 | 8712 | 0.276 | 1.36496 |
| Australia | 0.554664879 | 24,126 | 0.337 | 1.36162 |
| France | 0.518755819 | 64,721 | 0.295 | 1.37868 |
| New Zealand | 0.559156682 | 4661 | 0.349 | 1.3728 |
| Canada | 0.553760096 | 36,290 | 0.318 | 1.37564 |
| Finland | 0.52907638 | 5503 | 0.26 | 1.37403 |
| Luxembourg | 0.24632912 | 576 | 0.284 | 1.38491 |
| Denmark | 0.440207624 | 5712 | 0.256 | 1.36221 |
| Netherlands | 0.406882642 | 16,987 | 0.303 | 1.36485 |
| Sweden | 0.419781897 | 9838 | 0.278 | 1.36873 |
| Norway | 0.36105076 | 5255 | 0.272 | 1.35832 |

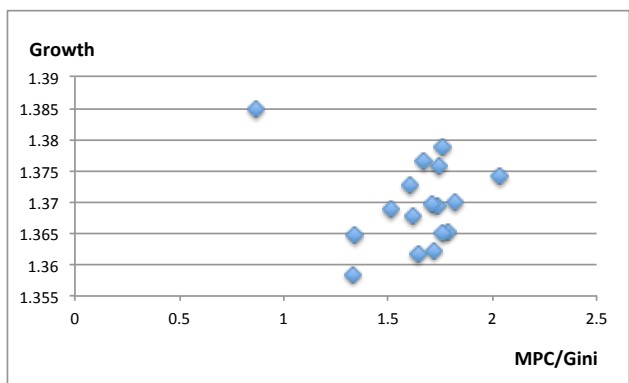

**Figure 12.** The diagram shows the relationship between the marginal propensity to consume/Gini coefficient and growth potential.

Next, we compared the growth potential of these countries with that of the United States by selecting two countries, namely, the United Kingdom and Japan, France and Switzerland, and Norway and Luxembourg, from among the countries with high, medium, and low marginal propensity to consume, respectively, and set the initial value of the Gini coefficient at 0.1 to 0.9. Figure 13 summarizes the results of the growth potential after the implementation of 3000 economic activities, with each initial Gini coefficient value ranging from 0.1 to 0.9 in increments of 0.01 for each of the following three cases. In the following figures (Figure 13), the growth potential of the US generally exceeds that of other countries in the interval, with a Gini coefficient of 0.4 or less because of the high marginal propensity to consume and the large population size. On the other hand, Norway and Luxembourg, where the marginal propensity to consume is relatively low and the population size is small, has a characteristic of maintaining stable growth in the interval with a highly unequal Gini coefficient of over 0.7 due to its structure that does not rely on consumption, in the same way as the interval with a high uniformity of income distribution.

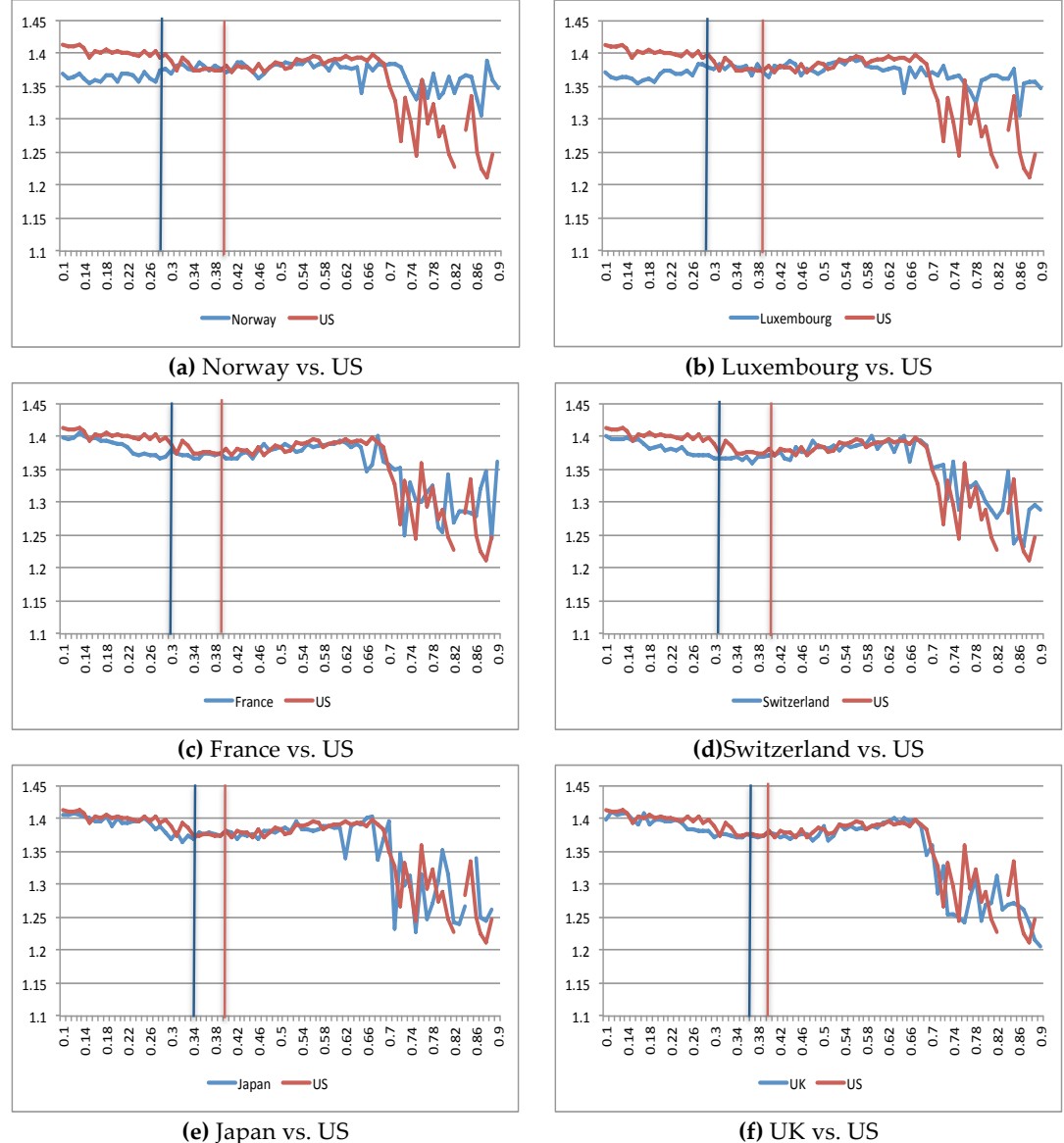

**(a)** Norway vs. US

**(b)** Luxembourg vs. US

**(c)** France vs. US

**(d)** Switzerland vs. US

**(e)** Japan vs. US

**(f)** UK vs. US

**Figure 13.** The diagrams show the simulations of growth potential in major OECD countries. Note: The horizontal axis shows the level of each Gini coefficient as the initial value. The vertical axis is the growth potential (calculation result/initial value of gross production). Vertical lines in the figure indicate the actual Gini coefficient levels in each country.

## 6. Discussion

### 6.1. Usefulness and Requirements of Agent-Based Simulation

In this chapter, let us discuss the simulation results referencing the literature that discusses the validity and significance of agent-based simulation methods and generated data.

This paper proposes a modeling technique for income distribution and long-term economic growth by applying agent-based simulation [57–60] rather than traditional mathematical analysis methods while conforming to the framework of orthodox macroeconomic models [15,16,34,35] and industrial organization theory on technological innovation [18–22]. This is because the macroeconomy has a complex structure that arises from its interrelationships with a large number of factors and because it is safe to say that it has not been able to provide practical policy theories on the policy issues above, both theoretically and empirically, especially on research that directly relates the two [48,50,53,55].

In general, in conducting agent-based simulations, the following requirements are presented by summarizing various research results on the relevant fields [50,55].

By agent-based simulations:

(1) We can produce results that are consistent with reality, although social phenomena, unlike natural phenomena, are not reproducible;

(2) we can reproduce in a limited sense a real phenomenon that is difficult to explain using existing mathematical theories;

(3) we can obtain satisfactory simulation results that are not to be arbitrary due to multiple parameter tunings;

(4) we can evaluate the validity of the results based on the applied theory, the basis of the implementation function, etc.; and

(5) we can approach problems that are difficult to explain using existing mathematical theories.

In this chapter, we will confirm the validity of the simulation results obtained in this paper based on the requirements mentioned above. Among the above requirements, let us first consider "consistency with reality" in (1). The model in this paper is built as an agent-based model, in the sense that the way results are derived uses simulations. The model structure itself is an orthodox macroeconomic model with an economic activity sector based on national accounts. On the other hand, as a method of decision-making, it adopts replicator dynamics that implements an adaptive behavior hypothesis (not a rational behavior hypothesis) corresponding to PDCA in the actual business field. As a result, a mathematical analysis is not possible, and instead, we attempt to generate data by simulation for non-reproducible real social phenomena. The population size, marginal propensity to consume, and Gini coefficient, which were adopted as conditional values for data generation, are all standard variables used in actual economic statistics and macroeconomics, and the generated simulation data are an extension of these observed data and are consistent with reality.

Next, let us consider "limited reproducibility" in (2). Due to the complexity of the target economic phenomena caused by multiple agents, the conventional macroeconomic modeling method needs to be simplified enough to obtain an analytical solution. As a result, it has been difficult to deal with actual problems [49]. On the other hand, ad-hoc case studies cannot ensure sufficient reliability to withstand wide and long-term policy recommendations. The model in this paper follows an orthodox macro-economic model and extends it to agent-based simulations to generate and reproduce data reflecting the actual conditions and characteristics of major OECD countries while dealing with measurable economic variables, such as the economic growth rate and Gini coefficient, concerning the problem of long-term economic growth and income distribution that has been difficult to solve until now. The simulation results of the growth potential of major OECD countries reproduce the data that the performance of the United States is currently higher than that of other OECD countries (see Section 5.3).

Let us consider the "arbitrariness due to multiple parameter tuning" in (3). The economic variables used in this paper are the population size, marginal propensity to consume, and Gini coefficient, all of which are adopted as standards in actual economic statistics and macroeconomics. The simulation in this paper uses actual values published in statistics, such as the OECD, or their estimates as the basis. On the other hand, the structural parameters defined in Equations (8) and (9) have been established based on Scherer's empirical work [64] to implement the sustained occurrence of technological innovation. The data generated from the simulations in this paper are consistent with these statistical and empirical studies, not to be arbitrary, and are satisfactory.

Consider the "results validation" in (4). The theoretical and practical reasons for constructing the agent-based model in this paper are as described in (1). In addition, simulation data are generated based on evidence, such as actual OECD statistics. Therefore, it is possible to evaluate the validity of simulation results on the basis of the applied theory in constructing the model, and the implemented function based on reality.

Finally, let us consider "access to issues that are difficult to explain in existing mathematical theories" in (5). Existing economic theories traditionally assume a kind of rationality that is very strict about agent behavior and decision-making [50]. However, not all strategies and policies are implemented based on the assumption of rationality in actual economic phenomena, especially in practical situations. With regard to policy issues related to equality in income distribution, behind the macroeconomic phenomenon that appears in the Gini coefficient is the fact that income levels and economic activities of individual citizens interact in a multidimensional and complex manner, and so it is unrealistic to assume a single rational agent in the national economy. This paper extends and expands the existing macroeconomic growth model by approximating actual practical economic activities through the multi-agent approach based on the setting of the economic activity sector and the adoption of an adaptive decision-making mechanism similar to PDCA. As a result, this paper attempts to systematically approach policy issues related to economic growth and income distribution, which have been difficult to explain until now, from data generated by simulation based on actual data on major OECD countries.

*6.2. Evidence-Based Policy Recommendations that Do Not Require Social Experiments*

In light of the above requirements, this paper makes feasible recommendations for policy options derived from agent-based simulations. In this paper, in conducting agent-based simulations, the characteristics specific to each country are defined by the levels of marginal propensity to consume, population size, and Gini coefficient, and given as initial conditions.

As a benchmark, we set a fixed population size of $10^8$ = 100 million people and set the initial marginal propensity to consume, with high as 0.91, medium as 0.5, and low as 0.091. The initial value of the Gini coefficient was set in increments of 0.01 from 0.01 to 0.99, and calculations corresponding to 3000 economic transactions were performed. As a result, 297 (3 × 99) types of inductively analyzable time-series data [54] on economic growth potential were generated.

Based on the above classification of marginal propensity to consume, we selected seven OECD countries, namely, the United States, United Kingdom, and Japan as high ranking countries, France and Switzerland as medium ranking countries, and Norway and Luxembourg as low ranking countries, and tried to generate time-series data on economic growth potential based on the actual conditions of each country (using real data). In addition, developing the initial Gini coefficient values in increments of 0.01 from 0.1 to 0.9, including actual measurements, provided a basis not only for forecasting economic growth potential based on current conditions but also for alternative policy options for maintaining or changing income distribution policies using the Gini coefficient as an alternative to social experiments. As a result, 567 (7 × 81) kinds of inductively analyzable time-series data [54] on economic growth potential were generated.

In this paper, we conduct agent-based simulations to generate 864 (297 + 567) types of inductively analyzable time-series data [54] on the above economic growth potential. By substituting time-series data related to individual circumstances in each country for social experiments as evidence [48,50,53,55], this paper provides a basis for practical examination based on the PDCA cycle concerning income distribution policy (induction to consumption demand-led type with increasing marginal propensity to consume and investment demand-led type with decreasing marginal propensity to consume) with the direction of technological innovation as an option. Below, we summarize the concrete results obtained from the model in this paper and expand on its policy implications. The policy recommendations by our agent-based simulations are as follows.

6.2.1. Policy Recommendation 1: High Growth Path with Equality or Unstable Path with Inequality in the Consumption-Led Economy (US, UK, and Japan)?

First, in an economy with a high marginal propensity to consume in the initial state, such as the USA, UK, and Japan, growth in aggregate consumption demand will be sustained by an equal distribution of gross income per capita, resulting in sustained relatively high growth through

consumption-driven knowledge stock selection over time. However, as the economy shifts to a more unequal income distribution state, its growth potential declines rapidly and it becomes highly volatile and highly unstable. Therefore, in countries with a high marginal propensity to consume currently, it is essential to maintain or shift income distribution with high equality.

6.2.2. Policy Recommendation 2: Stable and Moderate Growth Path even with Inequality in an Investment-Led Economy (Luxembourg and Norway)

Second, in an economy with a low marginal propensity to consume in the initial state, such as Luxembourg and Norway, even if gross income is distributed evenly per capita, the effect of sustaining the growth of gross consumer demand is small. As a result, sufficient growth cannot be achieved through consumption-led knowledge stock selection in comparison with the economy with a high marginal propensity to consume. However, even if economic inequality increases, the decline in growth potential is moderate and the dispersion is very small. The simulation results show that even in an economy with a high concentration of income, it is possible to maintain a stable society with investment-led growth where the growth potential fluctuates little, even though investment demand fluctuates unstably.

6.2.3. Policy Recommendation 3: Equalization of the Income Distribution When of the Generation of Technological Innovation

Third, I would like to mention the remaining and additional issues regarding the marginal propensity to consume. In general, the marginal propensity to consume can shift upward as new consumer goods markets emerge as technology advances. On the other hand, in the model presented in this paper, the marginal propensity to consume diminishes monotonously with an increase in income and does not have the structure that increases with the accumulation of knowledge stock. However, the results of the simulation can be expanded as follows. When the marginal propensity to consume rises due to the emergence of a new consumer goods market along with technological progress, even in all countries containing Luxembourg and Norway, higher growth potential can be realized by guiding policies to a society with a high degree of equality.

**Funding:** This research received no external funding.

**Conflicts of Interest:** The authors declare no conflict of interest.

**Appendix A**

In this paper, we discard the problem of the allocation between consumptions and savings at different points in time. Instead, we adopt the assumption that the consumer's marginal propensity to consume diminishes as income rises. The diminishing marginal propensity to consume is not a commonly adopted assumption. However, Keynes noted, "The marginal propensity to consume is not constant for all levels of employment, and it is probable that there will be, as a rule, a tendency for it to diminish as employment increases; when real income increases, that is to say, the community will wish to consume a gradually diminishing proportion of it" [65]. Actually, for the 26 OECD countries for which it is possible to estimate the marginal propensity to consume in each period, there is a negative relationship between the marginal propensity to consume and the GDP per capita, estimated in the both 1970 to 1989 and 1990 to 2016 periods, respectively, which is significant at less than 5% (less than 1% excluding South Korea, which has a slightly wider margin) and less than 0.1%, respectively.

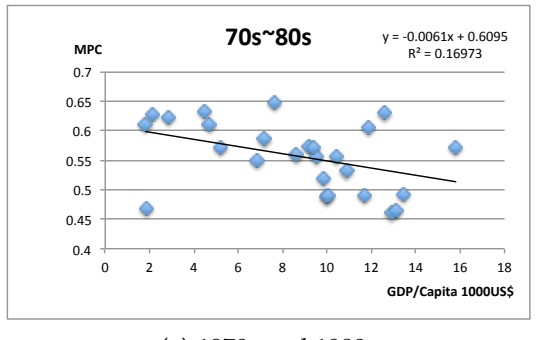 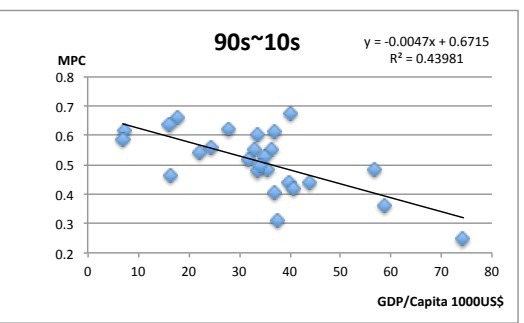

**(a)** 1970s and 1980s                **(b)** 1990s to 2010s

**Figure A1.** The diagrams show the relationship between GDP per capita and marginal propensity to consume in 26 OECD countries. Note: The 26 countries of OECD are Australia, Austria, Belgium, Canada, Denmark, Finland, France, Germany, Greece, Iceland, Ireland, Italy, Japan, Korea, Luxembourg, Mexico, the Netherlands, New Zealand, Norway, Portugal, Spain, Sweden, Switzerland, Turkey, the United Kingdom, and the United States. Source: United Nations and OECD.

On the other hand, in general, the marginal propensity to consume may increase with rising incomes if new products and services emerge that drive the macroeconomy through technological innovation. In this model, the knowledge stock, which is a production factor, is accumulated while the results of R & D fluctuate over time. Thus, since the model adopts an AK-type production function, its marginal productivity does not monotonously decrease but fluctuates. However, the associated mechanism of fluctuation in the marginal propensity to consume is not implemented, and the marginal propensity to consume diminishes monotonously with the increase in production, i.e., income. Although it is possible to construct a consumption function that implements a mechanism that varies with the marginal propensity to consume over time, we can approximately replace such a mechanism by comparing the calculation results with each other in the model, which starts from some cases of initial values of the marginal propensity to consume.

**Appendix B**

Generally, in the consumption function with diminishing marginal propensity to consume, the aggregate consumption amount in the perfectly even distribution type exceeds that in the perfectly concentrated distribution type in all domains with $Y > 0$. We now assume that the consumption function of diminishing marginal propensity to consume is as follows:

$$0 \leq C(Y),\ 0 \leq C'(Y) \leq 1,\ C''(Y) < 0\ for\ \forall Y \geq 0 \tag{A1}$$

$$C(0) = 0,\ C'(0) \leq 1 \tag{A2}$$

Based on the above assumptions, the function $F(Y)$ is defined as follows, where the population is $N \geq 1$ and we take the difference between the perfectly even distribution consumption function, $N\,C(Y/N)$, and the perfectly concentrated distribution consumption function, $C(Y)$:

$$F(Y) \equiv N\,C(Y/N) - C(Y)\ for\ \forall Y \geq 0 \tag{A3}$$

$$\Rightarrow F'(Y) = C'(Y/N) - C'(Y) \tag{A4}$$

First, when $N = 1$, the following relationship is established by the definition (A3). That is, there is no difference in the consumption function according to the distribution type:

$$F(Y) = 0\,for\ \forall Y \geq 0$$

Next, consider the case of $N > 1$.

(1)    In the case of $Y = 0$.
    The assumptions (A1) and (A2) and the definitions (A3) and (A4) make the following relationships:

$$F(0) = N\,C(0) - C(0) = 0 \ for \ \forall N > 1 \tag{A5}$$

$$F'(0) = C'(0) - C'(0) = 0 \ for \ \forall N > 1 \tag{A6}$$

(2)    In the case of $Y > 0$.
    The assumption (A1) makes the following relationships:

$$0 < Y/N < Y \ for \ \forall N > 1$$

$$\therefore F'(Y) = C'(Y/N) - C'(Y) > 0 \tag{A7}$$

Following the results of (A6) and (A7), $F(Y)$ is a monotonic increasing function for $\forall Y \geqq 0$ when $N > 1$. Therefore, in combination with the result of (A5), $F(Y) \geqq 0$ for $\forall Y \geqq 0$. As a result, the following relational equation (A8) holds. That is, in the consumption function in which the marginal propensity to consume diminishes, the aggregate consumption amount in the perfectly even distribution type exceeds that in the perfectly concentrated distribution type in all the definition areas where $Y > 0$:

$$0 \leq F(Y) \equiv N\,C(Y/N) - C(Y) \ for \ \forall Y \geq 0$$

$$\Rightarrow N\,C(Y/N) > C(Y) \ for \ \forall Y > 0 \tag{A8}$$

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
