# Peer review of "Equality in Income and Sustainability in Economic Growth: Agent-Based Simulations on OECD Data"

_sustainability, doi:10.3390/su11205803_

Round 1

Reviewer 1 Report

please find attached file

Author Response

Thank you for your review comments. This paper has been completely revised based on the following revised requirements. Because of the wide range of modifications, relevant parts in the paper are described in the "Blue Letter".

"The paper discusses the impact of income distribution on the macroeconomy by simulation using an agent-based model based on replicator dynamics. After a further round of an in-depth review by the authors, I would like to comment on the following academic weaknesses that should be addressed before any consideration for publication. Before any future submission for publication, however, it would be good for the authors to consider the following comments.

Recommendation:

More specifically,

Revision is requested before any attempt for publication will most likely require to take into account the following comments.

My general concern in that I believe that even authors made a major econometric effort and sit on relevant data, I struggle to find an original contribution in this paper."

→ The major correction was made as follows. By placing this paper in the context of existing research, the significance and uniqueness of the research were clarified (Chapter 1), the overall plan of this research was clarified (Chapter 2), and the results and policy significance of this research were revised to discuss along the evaluation procedure of general agent-based simulation research (Chapter 6).

"Other issues about this manuscript:

The title that is complex and does not reflect the content of the paper."

→ Title changed to " Equality in income and sustainability in economic growth: Agent-based simulations on OECD data."

"Abstract is far too long, dense and detailed, and it's not clear to understand how all the considerations are related to the main aim of the paper. Also, the innovation of the paper is missing in this part."

→ I rewrote it briefly.

"Almost all graphs have grammatical errors (for example there are a lot of question marks) that make very difficult to be understood."

→ The meaning of the grammatical error is unknown, but the graph in the text is pasted in PDF, and the original is uploaded separately as Supplementary files and figures, so please refer to it.

"Introduction: lines 64-90 present the differences from another paper (own publication). This part should be more compact. Also, it is needed to clearly explain how this paper will contribute to scientific research. For example, it is an empirical article; thus, why it is so important for scientific circles?"

→ I rewrote it briefly as such. Previous studies on economic growth theory, finance, and agent-based models were reviewed and this paper was placed in the context of these previous studies.

"Major changes in the conclusion part have to be done in order to make them relevant for policymakers and an international audience. For instance, please clearly provide your policy implications.

Authors should discuss which is the impact of their study findings upon society and state."

→ I have briefly described the points you pointed out and have rewritten them as a whole. The analytical methods and simulation results of this study were positioned as general requirements for the analytical methods based on agent-based models, and were completely rewritten to develop policy implications.

Reviewer 2 Report

The paper addresses a contemporary topic, still there are several issues that need to be strengthened and/or reviewed on the submitted version, namely:

Literature review misses several seminal works and important advances crossing distribution policies and economic growth. Presenting a review based on little more than 6 authors on such a rich domain is, naturally, short on a quality journal.

Material and methods presents several flaws, considering not only that it is hard to understand the used methodology, and thus to replicate the study, but also because the research steps are not adequately described. I would recommend the introduction of a phased methodological diagram.

The results are also very hard to read and there are several superfluous figures which add little or none to the presented information, e.g. Figures 6, 9, or 11... Further information is needed and assumptions need to be supported on further data.

Conclusions need to be more scientific. As they are they highlight the limitations of the research.

Author Response

Thank you for your review comments. This paper has been completely revised based on the following revised requirements. The major correction was made as follows. By placing this paper in the context of existing research, the significance and uniqueness of the research were clarified (Chapter 1), the overall plan of this research was clarified (Chapter 2), and the results and policy significance of this research were revised to discuss along the evaluation procedure of general agent-based simulation research (Chapter 6).

Because of the wide range of modifications, relevant parts in the paper are described in the "Blue Letter".

"The paper addresses a contemporary topic, still there are several issues that need to be strengthened and/or reviewed on the submitted version, namely:

Literature review misses several seminal works and important advances crossing distribution policies and economic growth. Presenting a review based on little more than 6 authors on such a rich domain is, naturally, short on a quality journal."

→ Previous studies on economic growth theory, finance, and agent-based models were reviewed and this paper was placed in the context of these previous studies.

"Material and methods presents several flaws, considering not only that it is hard to understand the used methodology, and thus to replicate the study, but also because the research steps are not adequately described. I would recommend the introduction of a phased methodological diagram."

→ I appreciate your proposal about the introduction of a phased methodological diagram. With reference to the overall picture of the model (Figure 3), the procedure for constructing the agent-based model employed in this study and the procedure for conducting the simulation have been completely rewritten. Table 1 summarizes the above procedures.

"The results are also very hard to read and there are several superfluous figures which add little or none to the presented information, e.g. Figures 6, 9, or 11... Further information is needed and assumptions need to be supported on further data."

→ Figure 6 has been deleted, and with regard to Figure 9 (Figure 8 in the revised version) and Figure 11 (Figure10 in the revised version), explanations regarding the derivation procedure have been added.

"Conclusions need to be more scientific. As they are they highlight the limitations of the research."

→ The analytical methods and simulation results of this study were positioned as general requirements for the analytical methods based on agent-based models, and were completely rewritten to develop policy implications.

Round 2

Reviewer 1 Report

Dear authors,

I would like to thank you for the opportunity I had to read your article again.
I can say that all required changes were done. However, my main concern that remains is that your article is a very long article that makes it  for any reader difficult to read and understand it.

Author Response

Thank you for your comments.

The author revised completely on typographical errors, grammar, syntax, and style throughout this paper. As a result, although the amount of text has not decreased, the paragraph structure in especially the discussion chapter, and wording have been reviewed and readability has improved.

Reviewer 2 Report

The introduced changes benefited a lot the overall quality of the paper.

A final typo and grammar review is mandatory.

Conclusions continue to be very broad. The conclusion chapter should highlight the strengths of the research emphasising the main findings.

Author Response

Thank you for your comments.
The author revised completely on typographical errors, grammar, syntax, and wording throughout this paper.
The structure of the discussion chapter was revised to highlight key conclusions as policy recommendations in the headlines.